# Rethinking Causal Mask Attention for Vision-Language Inference

**Xiaohuan Pei** [*]
Department of Computer Science
The University of Sydney, Australia
`xiaohuan.pei@sydney.edu.au`

**Tao Huang**
Department of Computer Science
Shanghai Jiao Tong University, China
`t.huang@sjtu.edu.cn`

**Yanxiang Ma**
Department of Computer Science
The University of Sydney, Australia
`yanxiang.ma@sydney.edu.au`

**Chang Xu** [†]
Department of Computer Science
The University of Sydney, Australia
`c.xu@sydney.edu.au`

## Abstract

Causal attention has become a foundational mechanism in autoregressive Vision-Language models (VLMs), unifying textual and visual inputs under a single generative framework. However, existing causal mask-based strategies are inherited from large language models (LLMs) where they are tailored for text-only decoding, and their adaptation to vision tokens is insufficiently addressed in the prefill stage. Strictly masking future positions for vision queries introduces overly rigid constraints, which hinder the model's ability to leverage future context that often contains essential semantic cues for accurate inference. In this work, we empirically investigate how different causal masking strategies affect vision-language inference and then propose a family of future-aware attentions tailored for this setting. We first empirically analyze the effect of previewing future tokens for vision queries and demonstrate that rigid masking undermines the model's capacity to capture useful contextual semantic representations. Based on these findings, we propose a lightweight attention family that aggregates future visual context into past representations via pooling, effectively preserving the autoregressive structure while enhancing cross-token dependencies. We evaluate a range of causal masks across diverse vision-language inference settings and show that selectively compressing future semantic context into past representations benefits the inference.

## 1 Introduction

In recent years, autoregressive large language models (LLMs) have achieved remarkable breakthroughs in linguistic understanding by enforcing causal attention mechanisms that restrict each token to attend only to its preceding context Achiam et al. (2023); Anil et al. (2023); Bai et al. (2023); Li et al. (2025); Radford et al. (2019). This left-to-right causal masking effectively prevents information leakage from future tokens, aligning model predictions with the natural sequential structure of language. For instance, in the sentence "She is very smart", predicting the token "very" based on the token "smart" would violate causality, as the model could also generate "not smart" given the same prefix. By restricting attention to past tokens, causal masking ensures consistent and contextually appropriate predictions.

Vision-language models (VLMs) Liu et al. (2023; 2024a); Bai et al. (2025); Chen et al. (2024b); Li et al. (2025); Chu et al. (2023), which extend LLMs to multi-modal settings, adopt a similar autoregressive framework by aligning and concatenating visual tokens with textual tokens. However, unlike text, visual information is inherently non-sequential, with regions processed holistically

---

[*]This work was supported in part by the Australian Research Council under Projects DP240101848 and FT230100549.

[†]Corresponding Author

rather than strictly in order. Consequently, enforcing strict causal masking on visual tokens may unnecessarily restrict the model's capacity to leverage contextual cues from future tokens. Recent studies Yin et al. (2024) suggest that future semantic attention scores, typically masked in causal settings, can be exploited without violating causal logic. Furthermore, Qi et al. (2025) indicates that visual tokens may not benefit as significantly from positional interactions, highlighting a potential misalignment between the causal structure designed for text and the optimal structure for visual processing.

As a result, a critical question arises: *Is causal attention truly a feasible mechanism for vision-language understanding?* In this paper, we systematically explore the impacts of causal attention on textual and visual tokens and reveal a surprising finding: while breaking the causal masks between textual tokens significantly disrupts model predictions, relaxing the causal constraints on visual tokens unexpectedly improves performance, even though the model is trained causally (see Figure 1).

To comprehensively investigate this phenomenon, we conduct an in-depth analysis of how relaxing future attention for visual tokens affects model behavior across diverse vision-language tasks, particularly those involving long contexts and multi-image reasoning. We propose three future-aware causal masking strategies, each targeting distinct regions in the multi-modal attention matrix. By examining their task-specific advantages and limitations, we uncover a variety of intriguing insights regarding the role of future visual context in enhancing inference accuracy. Our study aims to address the following key questions: *(1.) Does the causal attention in LLM fits the visual tokens in the popular VLMs like LLaVA? (2.) How should the causal attentions be revised to fit the multi-modal situation?*

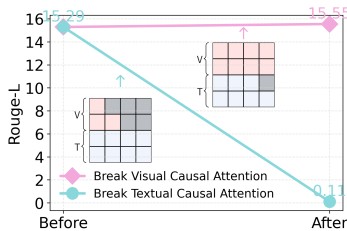

Figure 1: Breaking the casual masks of LLaVA-7b on the AL-FRED benchmark Shridhar et al. (2020).

*(3.) What tokens should the vision tokens be allowed to access in the causal mask? (4.) How does pre-seen visual semantic information impact tasks that heavily rely on visual reasoning? How about text-dependent tasks?*

Based on our findings, we conclude that allowing visual tokens to access future context significantly enhances VLM performance. However, directly breaking the causal masks between visual tokens substantially increases computational cost. To effectively incorporate future information while maintaining computational efficiency, we propose a kernel pooling method that merges future semantic attention into past regions. Additionally, we uncover several intriguing insights, such as the pronounced impact of merging future attention into attention sink regions, merging future into past even outperforms direct future access in some tasks, which notably alters VLM inference behavior.

As a result, this paper revisits the overlooked role of causal attention for visual tokens in VLMs and systematically investigates its limitations and alternatives. By proposing and evaluating a family of future-aware masks along with lightweight merging techniques, it offers both empirical gains and conceptual insights that challenge the default fundamental autoregressive design inherited from text-only LLMs.

## 2 PRELIMINARY

### 2.1 CAUSAL ATTENTION MECHANISM.

For vision-language models (VLMs), the input consists of $m$ visual tokens $X^v \in \mathbb{R}^{B \times m \times H \times D}$ and $n$ textual tokens $X^t \in \mathbb{R}^{B \times n \times H \times D}$, which are concatenated into a unified sequence $X = X^v \oplus X^t \in \mathbb{R}^{B \times L \times H \times D}, L = m + n$. The input $X$ is then projected into queries, keys, values: $Q, K, V \in \mathbb{R}^{B \times L \times H \times D}$. During the prefill stage, causal attention is computed as:

$$A = \text{Softmax}\left(\frac{QK^\top}{\sqrt{d}} + M^c\right)V, \quad M_{i,j}^c = \begin{cases} 0, & \text{if } j \leq i \\ -\infty, & \text{otherwise}, \end{cases} \quad (1)$$

where $A \in \mathbb{R}^{B \times H \times L \times D}$, $L = m + n$, with $m$ and $n$ being the number of visual and text tokens respectively. The mask $M \in \mathbb{R}^{L \times L}$ enforces autoregressive constraints across the entire sequence.

For each query token $i$, the causal masked row $M_i$ will be initialized as

$$M_i^c = [\underbrace{M_{i1}, M_{i2}, \cdots, M_{ii}}_{i \text{ (past)}}, \underbrace{-\infty, \cdots, -\infty}_{L-i \text{ (future)}}]_L, \tag{2}$$

## 2.2 Vision Language Model

Vision Language Models (VLMs) like LLaVA Liu et al. (2023) transfer the image input $X^v$ into vision tokens $\mathbf{x^v} \in R^{1,m}$ via a pretrained vision encoder $g(X)$, where $m$ is the number of vision tokens. The vision tokens are projected into text feature spaces, but contain the information from the images as

$$(x_1^v, x_2^v, ..., x_m^v) = \mathbf{x}^v = g(X^v). \tag{3}$$

In VLM, after the vision encoder, the image tokens are treated as if they were text tokens. Both image tokens $x^v$ and text tokens $x^t$ are input into an LLM $f_\phi$ in sequence. Denote token $i$ in the token sequence by $x_i$, when there are $m$ vision tokens and $n$ text tokens, VLMs can be generally defined as,

$$x_o = f_\phi(x_1^v, x_2^v, \ldots, x_m^v; x_1^t, x_2^t, ..., x_n^t) \tag{4}$$

where $x_o$ is the feature of the output token. In the LLM, the input tokens are sent into the causal attention layers, where the context feature between the tokens will significantly affect the prediction Yang et al. (2021); Pei et al. (2024). To be more precise, we denote the image and text token separately. Let $Q^v$, $Q^t$, $K^v$, and $K^t$ denote the queries and keys for the $x^v$ and $x^t$, respectively, we define $B(x^v, x^t) = \frac{(Q^v \oplus Q^t) \cdot (K^v \oplus K^t)^\top}{\sqrt{d}}$, where $\oplus$ is the concatenate function. Follow 1, in VLM, the softmax attention can be defined as,

$$h_\theta(\mathbf{x}^v, \mathbf{x}^t; M^c) = \text{Softmax}\left(B(\mathbf{x}^v, \mathbf{x}^t) + M^c\right). \tag{5}$$

Then the attention output can be redefined as $A = h_\theta(\mathbf{x}^v, \mathbf{x}^t, M^c) \cdot V$. The distribution of the prediction with causal attention in VLM can be formulated as

$$p_\theta(x_o = x \mid \mathbf{x}_{1:m}^v, \mathbf{x}_{1:n}^t) = \frac{\exp\left(e(x)^\top h_\theta(\mathbf{x}_{1:m}^v, \mathbf{x}_{1:n}^t; M^c)\right)}{\sum_{x'} \exp\left(e(x')^\top h_\theta(\mathbf{x}_{1:m}^v, \mathbf{x}_{1:n}^t; M^c)\right)}, \tag{6}$$

where $x\prime$ is the entire output vocabulary, $e(\cdot)$ is the vector in attention. Eq. 6 shows that the context information in visual semantics is learned between vision tokens. However, Eq. 3 shows that the context information of vision semantics is fixed into the vision tokens $\mathbf{x^v}$ by the pre-trained vision encoder $g(X)$. This means that the causal attention conflicts with the vision encoder in context information comprehension. Intuitively, we believe that in VLM, the image tokens have huge potential unrevealed by not applying the causal attention mechanism on the image tokens.

## 3 Understanding of Causal Attention

We aim to investigate and release the potential of future context in causal attention for vision-language models (VLMs). We begin by conducting an empirical study that examines how visual tokens interact with future tokens under various causal masking strategies. This analysis reveals that letting visual tokens open access to future context has the potential to improve reasoning performance. Motivated by these findings, we further propose a lightweight mechanism that enables the model to benefit from future visual signals without breaking the autoregressive structure. Figure 2 presents an overview of our investigation. The future-aware causal mask allows vision tokens to preview attention scores from future tokens and selectively compresses valid future attention into past positions to enhance efficiency while preserving autoregressive constraints.

### 3.1 Future Aware Causal Masks

The mainstream VLM backbone consists of a vision encoder to project the visual patches to visual tokens, and afterwards concatenate them with text tokens. Given a set of visual tokens $\mathbf{x^v} = \{x_1^v, x_2^v, \ldots, x_m^v\}$ and text tokens $\mathbf{x^t} = \{x_1^t, x_2^t, \ldots, x_n^t\}$, a flattened input sequence can be constructed as $\mathbf{X} = \mathbf{x^v} \oplus \mathbf{x^t}$. In this paper we focus on the case where vision tokens are entered before text tokens. The subscripts related to $x^v$ is $\mathbb{Z} \cap [1, m]$, and to $x^t$ is $\mathbb{Z} \cap [m+1, m+n]$, where $\mathbb{Z}$ is the set of integers. For simplicity, we denote $\mathbb{Z} \cap [1, m]$ by $\mathcal{V}$, and $\mathbb{Z} \cap [m+1, m+n]$ by $\mathcal{T}$.

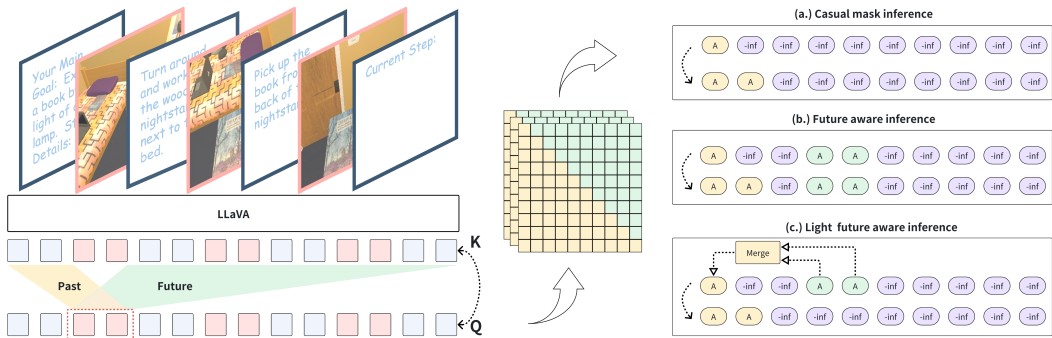

Figure 2: An overview of our investigation into causal attention in vision-language inference. **(a.) Casual mask inference**: enforces strict autoregressive decoding by blocking all future attention. **(b.) Future-aware inference:** enables visual tokens to preview future tokens in the upper-triangular region. **(c.) Light future-aware inference:** compresses future attentions into past visual positions.

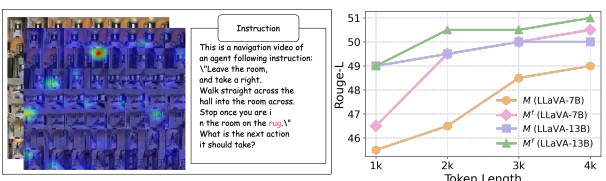

Figure 3: An Example of Temporal Multi-Images Task, Visual Navigation

Table 1: AP: Action Prediction Wu et al. (2024), VN: Visual Navigation Krantz et al. (2020), SC: State Change

| Mask | Temporal Multi-Image Tasks | | |
|------|------|------|------|
| | AP | VN | SC |
| $M$ | 39.8 | 31 | 30 |
| $M^f$ | 39.9($\uparrow$) | 32($\uparrow$) | 31.5($\uparrow$) |

This decomposition enables us to design future-aware variants of causal attention that selectively relax constraints for vision tokens while preserving strict autoregressive decoding for text. The standard design of causal attention prevents each token from attending to future positions and this constraint originally designed for text decoding, can be overly restrictive for visual tokens. To examine this, we propose a set of causal masking strategies that make the visual attention access the future sematic attention scores. As the future region of casual mask contains visual to visual ($v2v$) and visual to text ($v2t$), we define three future-aware variants mask strategy: Future-Aware Full Mask $M^f$, Future-Aware Visual-to-Visual Mask $M^{v2v}$ and Future-Aware Visual-to-Textual Mask $M^{v2t}$.

**Definition 3.1** (Future-Aware Full Mask). *For any query position $i \in \mathcal{V}$ (i.e., visual token), the future-aware full mask $M_i^f \in \mathbb{R}^L$ retains attention to all positions $j$, including future tokens in both visual and textual modalities:*

$$M_{i,j}^f = \begin{cases} 0, & \text{if } j \leq i \vee (j > i \wedge i \in \mathcal{V}) \\ -\infty, & \text{otherwise} \end{cases} \tag{7}$$

*Then the following holds:*

- *Full upper-triangle is visible for visual queries.*

- *Past causal structure is preserved: $M_{i,j} = 0$ for $j \leq i$.*

- *When $i \in \mathcal{T}$, standard causal mask is used.*

**Observation of $M^f$:** *Accessing full future attention scores for visual query could be beneficial to temporal multi-image tasks. Allowing visual tokens to attend to the entire future context enhances tasks that rely on global temporal reasoning, as it enables each visual query to incorporate upcoming visual attentions that be crucial for accurate inference and decision-making.*

**Analysis.** Figure 3 and Table 1 demonstrate that applying the full future-aware mask $M^f$ consistently improves performance across all temporal multi-image tasks. Specifically, on Visual Navigation (VN) and State Change (SC) tasks, which require long-horizon reasoning over temporally ordered image sequences, $M^f$ yields significant score gains over the standard causal mask. These tasks (e.g.

Egocentric Navigation, Action Sequence Prediction, and Scene Transition) demand the model to interpret actions or spatial arrangements over time. The full future mask allows each visual query to access all subsequent visual and textual context during the prefill stage. This enables the model to aggregate temporally rich semantics that are not yet locally visible but are crucial for understanding object motion trajectories, navigation goals, or state shifts. Such unrestricted future attention is particularly helpful in settings where key visual cues for inference (*e.g.*, an agent reaching a door or an object changing color) appear later in the image sequence. It reveals that the utility of full future-aware attention varies with task structure: it brings minimal benefit to static or single-image inputs but becomes indispensable for robust temporal modeling in multi-image, temporally structured settings.

**Definition 3.2** (Future-Aware Visual-to-Visual Mask). *For any query $i \in \mathcal{V}$, the visual-to-visual future mask $M_i^{v2v}$ permits attending to future visual tokens but masks future text tokens:*

$$M_{i,j}^{v2v} = \begin{cases} 0, & \text{if } j \leq i \vee (j > i \wedge i, j \in \mathcal{V}) \\ -\infty, & \text{otherwise} \end{cases} \tag{8}$$

*Then the following holds:*

- *Only future visual tokens are accessible to visual queries.*

- *Future text tokens are masked with $-\infty$.*

- *When $i \in \mathcal{T}$, reverts to standard causal masking.*

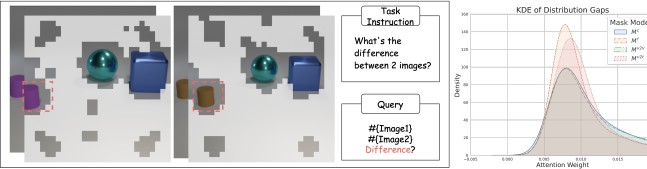

Figure 4: Example of a visual-relational task. The model must reason over relationships among visual tokens rather than isolated appearances.

Table 2: VCC: Visual Change Caption Jhamtani & Berg-Kirkpatrick (2018). VRE: Visual Relation Expression Hosseinzadeh & Wang (2021).

| Mask | Visual Relation Tasks | |
|---|---|---|
| | VCC | VRE |
| $M$ | 16.2 | 16.6 |
| $M^{v2v}$ | 16.7($\uparrow$) | 18.1($\uparrow$) |

**Observation of $M^{v2v}$**: *Allowing access to future visual tokens can benefit Visual Relation Inference tasks (e.g., Visual Change Captioning, Visual Relationship Expression), as it enables visual queries to capture interactions with future visual content—an essential component of reasoning about visual relationships.*

**Analysis.** Figure 4 and Table 2 show that applying the visual-to-visual future-aware mask $M^{v2v}$ leads to noticeable improvements on visual relation tasks such as Visual Change Captioning (VCC) and Visual Relation Expression (VRE). These tasks involve identifying subtle differences or relationships between two related images, where the visual context is rich but the textual signal is limited. By allowing visual queries to access future visual tokens during the prefill stage, $M^{v2v}$ enables the model to better compare visual patches across frames and capture object interactions or appearance changes. As illustrated in the distribution gap, the attention distribution under $M^{v2v}$ closely aligns with the original softmax distribution, indicating that this selective relaxation of the mask preserves natural attention behavior. The empirical results show that visual relational reasoning relies mainly on intra-modal alignment rather than cross-modal fusion. It benefits from accessing future visual tokens while textual tokens remain strictly causal.

**Definition 3.3** (Future-Aware Visual-to-Textual Mask). *For any query $i \in \mathcal{V}$, the visual-to-textual future mask $M_i^{v2t}$ allows access to future text tokens while masking future visual tokens:*

$$M_{i,j}^{v2t} = \begin{cases} 0, & \text{if } j \leq i \vee (j > i \wedge i \in \mathcal{V}, j \in \mathcal{T}) \\ -\infty, & \text{otherwise} \end{cases} \tag{9}$$

*Then the following holds:*

- *Visual queries could preview future textual attention scores.*

- *Future visual context is strictly masked.*

- *When $i \in \mathcal{T}$, attention follows standard left-to-right causality.*

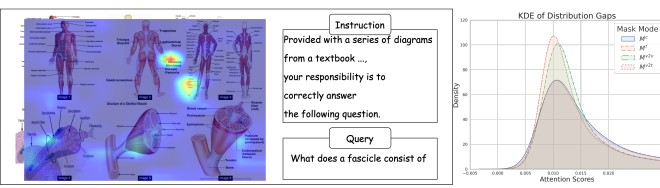

Figure 5: An Example of Text-Rich VQA Tasks

Table 3: Text-Rich Image QA Tasks: OCR-VQA Mishra et al. (2019) and TextVQA Kembhavi et al. (2017).

| Mask | Text-Rich Image QA Tasks | |
|------|------|------|
| | OCR-VQA | TextVQA |
| $M$ | 22.5 | 32.0 |
| $M^{v2t}$ | 23.0($\uparrow$) | 38.5($\uparrow$) |

**Observation of $M^{v2t}$:** *Enabling future access from visual tokens to textual tokens benefits Text-Rich Image QA tasks, as it allows visual queries to anticipate and integrate critical textual cues embedded in images—often the key to accurate reasoning and answer generation.*

**Analysis.** Figure 5 and Table 3 show that the visual-to-textual future-aware mask $M^{v2t}$ yields notable improvements in Text-Rich Image QA tasks such as OCR-VQA Mishra et al. (2019) and TextVQA Kembhavi et al. (2017). These benchmarks require extracting fine-grained textual information embedded in complex visual layouts—such as textbook diagrams or document images—where visual cues often need to resolve or align with distant text regions. By allowing visual queries to attend to future textual tokens, $M^{v2t}$ enables earlier visual patches to preemptively integrate relevant linguistic content during prefill, improving semantic alignment and grounding. Specifically, this attention mode avoids exposing future visual context, maintaining temporal consistency. The KDE distribution gap further indicates that the attention distribution under $M^{v2t}$ is better aligned with the natural softmax pattern than other variants, supporting the hypothesis that selective cross-modal future access can improve answer accuracy in scenarios dominated by image-embedded text.

Based on our definition in Def. 3.1, 3.2, and 3.3, the distribution of $x_o$ in Eq. 6 can be revised as

$$p_\theta(X_a = x \mid \mathbf{x}_{1:m}^v, \mathbf{x}_{1:n}^t) = \frac{\exp\left(e(x)^\top h_\theta(\mathbf{x}_{1:m}^v, \mathbf{x}_{1:n}^t; \mu)\right)}{\sum_{x'} \exp\left(e(x')^\top h_\theta(\mathbf{x}_{1:m}^v, \mathbf{x}_{1:n}^t; \mu)\right)}, \tag{10}$$

where $\mu$ is the modified mask strategies and $\mu \in \{M^{v2v}, M^{v2t}, M^f\}$ are selected manually and fixed.

## 4 LIGHT FUTURE AWARE ATTENTION FAMILY

While granting visual tokens access to future context holds great potential for improving multimodal understanding, such full visibility comes at the cost of increased inference latency—particularly during the autoregressive decoding phase. Fortunately, the recent trend of separating prefill and decoding stages in VLMs allows us to shift this overhead entirely into the prefill phase. Leveraging this separation, we propose a lightweight attention mechanism that compresses future visual information into past positions during prefill, enabling the model to benefit from future-aware context while preserving the original causal mask structure during decoding.

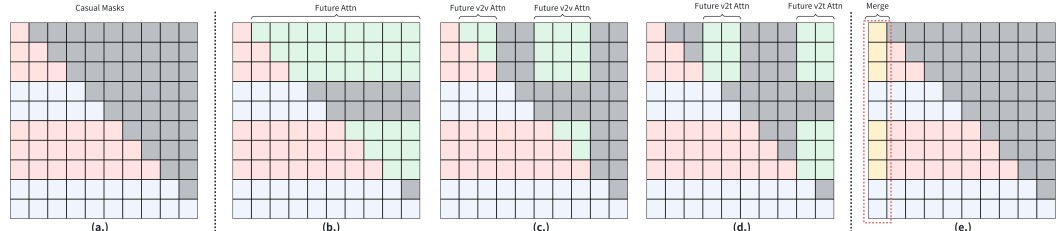

Figure 6: An overview of attention design for vision language inference. **(a.)** Casual Mask Attention. **(b.)** Future-Aware Full Attention. **(c.)** Future-Aware Visual-to-Visual Attention. **(d.)** Future-Aware Visual-to-Textual Attention. **(e.)** Light Future Aware Attention.

Table 4: Performance comparison across vision-language tasks using different future-aware causal masking strategies for visual queries. We evaluate the baseline causal mask ($M^c$), three future-relaxed variants ($M^{v2v}$, $M^{v2t}$, $M^f$), and their lightweight merge variants (prefix size = 1).

| Method | ActionL | ActionP | ActionS | CLEVR | Order | DocVQA | Nav | Moving | OCRVQA | Object | SpotDiff | State | TQA |
|---|---|---|---|---|---|---|---|---|---|---|---|---|---|
| | | | | | | LLaVA-7b | | | | | | | |
| $M^c$ | 0.230 | 0.515 | 0.445 | 0.166 | 0.245 | **0.450** | 0.310 | 0.490 | 0.225 | 0.485 | 0.162 | 0.300 | 0.320 |
| $M^{v2t}$ | 0.250 | 0.495 | 0.435 | 0.181 | 0.250 | 0.445 | 0.320 | 0.490 | **0.230** | 0.495 | 0.165 | 0.305 | 0.385 |
| $M^{v2v}$ | **0.255** | **0.515** | 0.440 | 0.177 | 0.250 | 0.430 | **0.325** | **0.515** | 0.220 | 0.500 | 0.167 | **0.325** | 0.385 |
| $M^f$ | 0.250 | 0.500 | **0.450** | 0.187 | 0.255 | 0.430 | 0.320 | 0.505 | 0.225 | 0.505 | 0.171 | 0.315 | **0.400** |
| $M^{v2v}$+merge | 0.225 | **0.51** | 0.435 | 0.175 | **0.27** | 0.445 | 0.320 | 0.490 | 0.205 | **0.510** | 0.167 | 0.305 | 0.385 |
| $M^{v2t}$+merge | 0.245 | 0.495 | 0.435 | 0.180 | 0.250 | 0.445 | 0.320 | 0.490 | 0.230 | 0.495 | 0.164 | 0.305 | 0.375 |
| $M^f$+merge | 0.245 | 0.500 | 0.450 | **0.188** | 0.265 | 0.420 | 0.320 | 0.505 | 0.225 | 0.490 | **0.173** | 0.320 | 0.375 |
| | | | | | | LLaVA-13b | | | | | | | |
| $M^c$ | 0.230 | 0.450 | 0.450 | 0.157 | 0.435 | 0.455 | 0.260 | 0.500 | **0.455** | 0.470 | **0.158** | 0.360 | 0.495 |
| $M^{v2t}$ | **0.245** | 0.455 | 0.495 | 0.156 | **0.445** | 0.460 | **0.270** | 0.500 | 0.415 | **0.475** | 0.120 | 0.360 | 0.515 |
| $M^{v2v}$ | 0.225 | 0.455 | **0.500** | 0.157 | 0.435 | **0.465** | 0.265 | 0.500 | 0.415 | **0.475** | 0.143 | 0.360 | **0.525** |
| $M^f$ | **0.245** | **0.460** | 0.495 | 0.156 | 0.440 | 0.460 | 0.260 | **0.510** | 0.415 | **0.475** | 0.155 | **0.370** | 0.510 |
| $M^{v2v}$+merge | 0.245 | 0.455 | 0.495 | 0.155 | 0.445 | 0.46 | 0.270 | 0.500 | 0.415 | 0.475 | 0.141 | 0.36 | 0.515 |
| $M^{v2t}$+merge | 0.245 | **0.455** | 0.495 | 0.155 | 0.445 | 0.46 | 0.270 | 0.500 | 0.415 | 0.475 | 0.119 | 0.360 | 0.510 |
| $M^f$+merge | 0.255 | 0.450 | 0.495 | **0.158** | 0.445 | **0.465** | 0.260 | 0.505 | 0.415 | **0.480** | 0.115 | 0.355 | 0.525 |

Motivated by the attention sink phenomenon observed in autoregressive models Gu et al. (2024); Xiao et al. (2024); Yang et al. (2024) and its effectiveness in recent inference optimization studies Ge et al. (2023); Liu et al. (2024b), we merge the compressed future information into the initial vertical past positions to enhance semantic propagation during prefill. We apply 1D kernel pooling over the attention weights using a kernel size $k$ to aggregate visual semantics, and merge the resulting summary score back into the past region $j \leq i$ of the same row $A_i$ as:

$$M_{i,j}^p(\mu) = \begin{cases} 0, & \text{if } j \leq i \text{ or } \mu_{i,j} = -\infty \\ 1, & \text{otherwise,} \end{cases} \tag{11}$$

$$C(B,\mu) = \begin{cases} \sum_{s=1}^{T-k+1} \max_{t=0}^{k-1} (B \odot M^p(\mu))_{i,i+s+t}, & \text{where } j \leq i \text{ and } j = 1 \\ 0, & \text{otherwise} \end{cases} \tag{12}$$

where $A_i \in \mathbb{R}^L$ denotes the attention distribution for query $Q_i$, $A_i^f$ represents its masked future segment, $k$ is the kernel size, $T = L - i - 1$ defines the maximum pooling range, $A^p$ is the aggregated semantic score via kernel pooling, and $C(B,\mu)$ is the attention row after merging. Then, the autoregressive generation process refines the token distribution from both the compressed future and the original past attention, producing predictions conditioned on enriched context representations:

$$h'_\theta(\mathbf{x}^v, \mathbf{x}^t; \mu) = \big(B(\mathbf{x}^v, \mathbf{x}^t) + C(B,\mu) + M^c\big) \tag{13}$$

Based on our definition in Def. 3.1, 3.2, and 3.3, the distribution of $x_o$ in Eq. 6 can be revised as

$$p_\theta(X_a = x \mid \mathbf{x}_{1:m}^v, \mathbf{x}_{1:n}^t) = \frac{\exp\big(e(x)^\top h'_\theta(\mathbf{x}_{1:m}^v, \mathbf{x}_{1:n}^t; \mu)\big)}{\sum_{x'} \exp\big(e(x')^\top h'_\theta(\mathbf{x}_{1:m}^v, \mathbf{x}_{1:n}^t; \mu)\big)}, \tag{14}$$

where $\mu \in \{M^{v2v}, M^{v2t}, M^f\}$ is selected manually and fixed, and $h'_\theta(\mathbf{x}_{1:m}^v, \mathbf{x}_{1:n}^t; \mu)$ is the modified mask attention family equipped with merged semantic future attentions.

The compressed method ensures that the final attention pattern remains strictly causal (lower-triangular) while still benefiting from future visual semantics aggregated during prefill.

**Analysis of Lightweight Attention Results.** Table 4 shows that the proposed lightweight attention strategy, which merges compressed future scores into a fixed prefix token, achieves competitive performance while preserving the standard causal structure. Across both 7B and 13B models, future-aware masks with prefix merging (such as $M^f$+merge and $M^{v2v}$+merge) perform on par with or better than their unmerged counterparts on tasks involving temporal reasoning, visual relations, and text-rich understanding. This indicates that full access to future tokens is not always necessary during decoding. Instead, summarizing future information into a small prefix, even a single token, provides sufficient global context for accurate generation.

Table 5: Effectiveness of different future-aware masking strategies . ✓: Consistent performance improvement across all benchmarks for the task. ✗: Performance degradation across all benchmarks. "-": Mixed results, some benchmarks improve while others degrade.

| Mask | Temporal Multi-Image Tasks | | | | Semantic Multi-Image Tasks | | | | | Needle In a Haytack | |
|------|------|------|------|------|------|------|------|------|------|------|------|
| | T-1 | T-2 | T-3 | T-4 | S-1 | S-2 | S-3 | S-4 | S-5 | N-1 | N-2 |
| M | ✓ | ✓ | ✓ | ✓ | ✓ | - | ✗ | ✓ | ✗ | - | - |
| $M^{v2v}$ | ✓ | ✓ | ✓ | - | - | - | ✓ | - | - | - | - |
| $M^{v2t}$ | - | - | - | - | - | ✓ | - | - | ✗ | - | - |

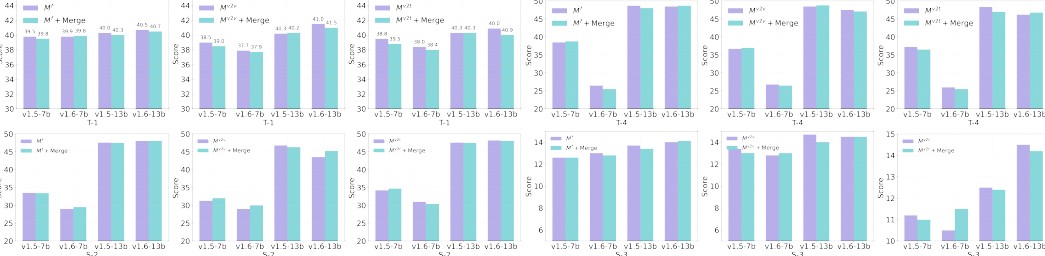

Figure 7: Performance comparison of three causal masks ($M^f$, $M^{v2v}$, $M^{v2t}$) and their lightweight merged variants across different model architectures (v1.5-Vicuna, v1.6-Mistral) and sizes (7b, 13b).

## 5 DISCUSSION

In this section, we address the questions posed in Section 1 through empirical analysis and experimental results. We further provide insights into these issues and explain how future-aware semantic design can support the development of vision-language models (VLMs).

**1. Causal attention from LLMs may not align well with vision tokens in VLMs and limits their contextual capacity.** Table 5 shows that relaxing the standard causal mask with future-aware strategies (Definitions 3.1, 3.2, 3.3) yields selective improvements across benchmarks, rather than uniform gains. Temporal multi-image tasks (T-1 to T-4) consistently benefit from $M^f$ and $M^{v2v}$, likely because they require modeling event sequences, spatial localization, and counterfactual changes over time. In these tasks, allowing visual queries to access future visual cues helps encode scene dynamics and long-term dependencies. Similarly, visual-relation tasks like S-2 and S-3 show gains under $M^{v2v}$ and $M^{v2t}$, suggesting that self-attention among visual tokens (*e.g.*, for spotting subtle differences) or previewing embedded text (*e.g.*, for reading labels in diagrams) enhances fine-grained reasoning. However, on text-dominant tasks (S-5, IR), these relaxed masks often degrade performance, confirming that strict autoregressive masking remains essential for textual alignment and matching.

**2. Causal attention could be revised by selectively relaxing future masking for vision tokens.** Definitions 3.1, 3.2, 3.3 introduce new masking strategies that modify the upper-triangular part of the causal mask. Instead of blocking all future tokens, these strategies allow visual queries to access selected future tokens: $M^f$ keeps all future tokens visible, $M^{v2v}$ keeps only future visual tokens, and $M^{v2t}$ keeps only future text tokens. As shown in Table 5, these changes improve performance in tasks that involve visual reasoning or temporal understanding. The results suggest that strict causal masking, designed for text, may be too limiting for vision. Allowing future attention in a controlled way helps vision tokens gather important context early, and better aligns the attention pattern with how visual information is structured.

**3. Vision tokens could attend to either or both visual and textual tokens based on task needs.** The three masking strategies defined in Definitions 3.1, 3.2, and 3.3 specify which types of future tokens visual queries may attend to. The experimental results in Table 4 and Figure 7 reveal that the optimal access pattern depends on the nature of the task. For visual relation inference (*e.g.*, visual change caption, visual rela- tion expression), $M^{v2v}$ performs best, as reasoning relies on capturing spatial or temporal relationships between future visual observations. In contrast, text-centric tasks like OCR-VQA and TextVQA benefit more from $M^{v2t}$, where visual tokens preview future textual content to interpret embedded text. Meanwhile, $M^f$ enables broad access to both modalities and helps in temporally grounded multi-image tasks. These findings suggest that causal masking could be

Table 6: Comparison of future-aware attention strategies with and without merging. Prefill Valid Attentions counts non-masked attention scores. $L$: The length of the prefill attention. $m/n$: the number of visual/textual tokens.

| Attention Type | Prefill Valid Attentions | Decoding Latency |
|---|---|---|
| $M^f$ | $L(L+1)/2 + mL - m(m+1)/2$ | 83.1783 ms/token |
| $M^f$+merge | $L(L+1)/2$ | 26.5362 ms/token |
| $M^{v2v}$ | $L(L+1)/2 + m(m-1)/2$ | 64.1266 ms/token |
| $M^{v2v}$+merge | $L(L+1)/2$ | 26.4037 ms/token |
| $M^{v2t}$ | $L(L+1)/2 + m \cdot n$ | 43.0362 ms/token |
| $M^{v2t}$+merge | $L(L+1)/2$ | 26.1051 ms/token |

flexibly adapted: vision tokens could be granted selective access to future visual or textual information based on the modality relevance of the downstream task.

**4. Pre-seen visual semantics show task-dependent benefits.** Allowing visual tokens to preview future content helps in tasks that rely heavily on intra-visual reasoning (e.g., Visual Change Captioning and Visual Relation Expression), while future text access proves more beneficial for text-dominant tasks (e.g., OCR-VQA, TextVQA). As shown in Table 2 and Figure 4, relaxing visual-to-visual constraints using $M^{v2v}$ leads to notable gains in visual relation benchmarks, where understanding visual relationships across multiple frames or regions is essential. Conversely, Table 3 and Figure 5 demonstrate that $M^{v2t}$ significantly boosts performance in text-rich visual QA tasks by letting visual tokens access future text cues early in the decoding process.

Building on the findings from future-aware masking strategies, we further provide some insights on the method of merging future attention into the past region.

**1. Merging future attention in the prefill stage could enjoy a trade-off of performance and latency in the latter decoding stage.** Figure 7 shows that merging pooled future attention into early prefix tokens retains most of the performance benefits offered by future-aware masks. Meanwhile, Table 6 quantitatively demonstrates that merging significantly reduces decoding latency (*As breaking the structure precludes computing attention scores via concatenation over past keys/values, we evaluate performance without KV cache for the no-merge variant.* ). Compared to the unmerged versions, applying merge leads to a reduction from 83.18 ms/token ($M^f$) to 26.53 ms/token ($M^f$+merge), from 64.13 ms/token ($M^{v2v}$) to 26.40 ms/token ($M^{v2v}$+merge), and from 43.04 ms/token ($M^{v2t}$) to 26.10 ms/token ($M^{v2t}$+merge). The 2-3× speedup stems from the fact that merged models rely solely on standard causal decoding, avoiding the overhead of computing extra future attention.

**2. Future semantics can be utilized by merging them into attention sink regions in the past.** To evaluate this, we define the *prefix ratio* as *prefix size*$/L$, where $L$ is the total attention length, and the *prefix size* refers to the number of past tokens into which the pooled future attention scores are merged. Figure 8 shows that as the prefix ratio increases, attention to future tokens decreases, indicating that future information can be compressed into earlier tokens through pooling. This preserves the autoregressive structure while enabling the model to access future semantics indirectly. The prefix acts as an attention sink that gathers and retains useful signals for subsequent genera-

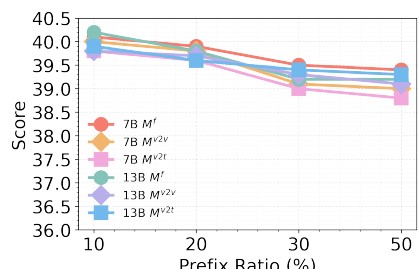

Figure 8: Effect of prefix ratio of Light future aware attentions.

tion. Interestingly, we find that merging the pooled future scores into just the first token already leads to strong results, suggesting that a single well-positioned sink token is often sufficient to absorb and propagate future context effectively.

## 6 CONCLUSION

In this work, we revisit causal attention in vision-language models (VLMs) and show that the standard left-to-right masking used in language models often misaligns with the characteristics of visual inputs.

We conduct a detailed empirical study across 15 multimodal tasks and propose three future-aware causal masking strategies that selectively expose future tokens to visual queries. These strategies lead to clear improvements on tasks requiring temporal, relational, or text-based reasoning. We also introduce a lightweight attention mechanism that compresses future attention into prefix tokens during prefill, preserving decoding efficiency while enhancing context modeling. We further analyze the root cause of the misalignment and provide insights that improve the understanding and design of modality-aware causal attention in vision-language models.

## ETHICS STATEMENT

This work studies inference-time causal masking for decoder-only vision–language models. The research does not involve human subjects, annotation efforts, or newly collected data. All tasks used in our evaluation come directly from *MILEBench*, a publicly available long-context multimodal benchmark suite. The benchmark aggregates existing datasets, including VQAv2, GQA, TextVQA, OCRBench, and Spot-the-Diff, under unified evaluation protocols. We use these datasets strictly under their respective licenses, and to our knowledge, none contain sensitive personal information. Our method modifies only the inference-time attention mask of open-source pretrained models without introducing new training or supervision. We encourage responsible and privacy-aware usage of this work and discourage applications that violate safety, fairness, or ethical norms.

## REPRODUCIBILITY STATEMENT

We aim to make this study fully reproducible. The proposed masking mechanisms, including $M^f$, $M^{v2v}$, $M^{v2t}$, merge-based variants, and baseline causal masks, are defined in the main text and appendix. All experiments rely on public *MILEBench* tasks, which provide standardized long-context and multimodal evaluation settings consolidated from VQAv2, GQA, TextVQA, OCRBench, and Spot-the-Diff. Dataset preprocessing, multimodal tokenization, prompt formats, and evaluation metrics are specified in the main paper.

All models are evaluated using the Hugging Face `transformers` library version $4.34.7$, which is compatible with `CUDA` $12.4$ and avoids conflicts with `FlashAttention` kernels. We use publicly released checkpoints of LLaVA-7B and LLaVA-13B without any additional training. Hyperparameters such as decoding settings, maximum context length, and mask-application rules remain fixed across all experiments and are reported in the appendix. Since our method is inference-only and preserves the original architecture, all results can be reproduced by applying our mask implementation to the released checkpoints. We will release our full implementation, configuration files, and scripts upon publication to support independent verification.

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

# A Technical Appendices and Supplementary Material

## A.1 Experimental Setup

All experiments were conducted on NVIDIA A100 GPUs using the official implementation of LLaVA series Liu et al. (2024a), with FlashAttention-2.6.3 [1] integrated for efficient attention computation. The context length was set to 4096 tokens, and all generations were performed using greedy decoding with a fixed temperature of 0 to ensure deterministic outputs. Task definitions and groupings follow the standard taxonomy established by MILEBench Song et al. (2024), covering a diverse spectrum of 29 multimodal benchmarks. To accommodate the memory and sequence length variability across datasets, batch sizes were dynamically adjusted: a batch size of 1 was used for long-context datasets such as MMCoQA Li et al. (2022) and GPR1200 Schall et al. (2022), while a batch size of 24 was adopted for the remaining tasks. For tasks with highly imbalanced attention patterns, we applied kernel-based attention merging strategies with top-k region ratios calibrated per dataset. We further incorporated minor task-specific biases—for example, a fixed bias of 0.5 in EgocentricNavigation Krantz et al. (2020) and 1.5 in SlideVQA Tanaka et al. (2023)—while retaining default configurations elsewhere. All preprocessing and evaluation followed the official MILEBench protocol to ensure fair and reproducible comparisons.

Table 7: Detailed Tasks inherited from MILEBench Song et al. (2024).

| Category | Task | Dataset | Data Source | Count | Metric |
|---|---|---|---|---|---|
| Temporal Multi-image | Action Understanding and Prediction (T-1) | Action Localization
Action Prediction
Action Sequence | STA Gao et al. (2017)
STAR Wu et al. (2024)
STAR Wu et al. (2024) | 200 | Accuracy |
| | Object and Scene Understanding (T-2) | Object Existence
Object Interaction
Moving Attribute
Object Shuffle | CLEVRER Yi et al. (2019)
STAR Wu et al. (2024)
CLEVRER Yi et al. (2019)
Perception Test Patraucean et al. (2024) | 200 | Accuracy |
| | Visual Navigation and Spatial Localization (T-3) | Egocentric Navigation
Moving Direction | VLN-CE Krantz et al. (2020)
CLEVRER Yi et al. (2019) | 200 | Accuracy |
| | Counterfactual Reasoning and State Change (T-4) | Counterfactual Inference
State Change
Character Order
Scene Transition | CLEVRER Yi et al. (2019)
Perception Test Patraucean et al. (2024)
Perception Test Patraucean et al. (2024)
MovieNet Huang et al. (2020) | 200 | Accuracy |
| Semantic Multi-image | Knowledge Grounded QA (S-1) | Webpage QA
Textbook QA
Complex Multimodal QA
Long Text with Images QA | WebQA Chang et al. (2022)
TQA Kembhavi et al. (2017)
MultiModalQA Talmor et al. (2021)
WikiVQA | 200 | Accuracy |
| | Text-Rich Images QA (S-2) | Slide QA
OCR QA
Document QA | SlideVQA Tanaka et al. (2023)
OCR-VQA Mishra et al. (2019)
DocVQA Mathew et al. (2021) | 200 | Accuracy |
| | Visual Relation Inference (S-3) | Visual Change Captioning
Visual Relationship Expressing | Spot-the-Diff Jhamtani & Berg-Kirkpatrick (2018)
CLEVR-Change Hosseinzadeh & Wang (2021) | 200 | ROUGE-L |
| | Dialogue (S-4) | Multimodal Dialogue
Conversational Embodied Dialogue | MMCoQA Li et al. (2022)
ALFRED Shridhar et al. (2020) | 200 | Accuracy |
| | Space Understanding (S-5) | nuScenes | nuScenes Caesar et al. (2020) | 200 | Accuracy |
| Diagnostic Evaluation | Needle In A Haystack (N-1) | Text Needle In A Haystack | TextNeedleInAHaystack | 320 | Accuracy |
| | Needle In A Haystack (N-2) | Image Needle In A Haystack | ImageNeedleInAHaystack | 320 | Accuracy |
| | Image Retrieval (I-1) | Image Retrieval | GPR1200 Schall et al. (2022) | 600 | Accuracy |

## A.2 Related Work

With the success of decoder-only large language models (LLMs) Achiam et al. (2023); Bai et al. (2023); Li et al. (2025); Touvron et al. (2023); Abdin et al. (2024), recent advances have extended their capabilities to the multimodal domain, giving rise to Vision-Language Models (VLMs). Early frameworks such as LLaVA Liu et al. (2023), InternVL Chen et al. (2024b), and Qwen-VL Bai et al. (2025) demonstrate that instruction tuning can be adapted to handle flatted textual and visual inputs, enabling strong performance across tasks such as visual reasoning, captioning, and instruction following. Additionally, most multi-modality pre-trained work Lin et al. (2024); Bai et al. (2025); Li et al. (2023b; 2024); Zou et al. (2024); Yang et al. (2025) also inherit the causal masking design from LLMs, which, while crucial for token generation, may unnecessarily constrain specific modality token (e.g. In our paper it refers to visual tokens). These pre-trained models typically flatten visual and textual tokens into a single sequence and feed them into an decoder-only, which may overlooking modality-specific attention patterns. To mitigate these limitations, recent studies have explored resolution-aware vision encoders Chai et al. (2022); Chen et al. (2024a), multi-modal alignment modules Singh et al. (2022), and fine-grained token interaction strategies Yao et al. (2021), aiming to better adapt LLM-based decoder-only architecture to the visual perception reasoning. And the potential usage of the future tokens has been shown in LLMs architecture Yin et al. (2024). Beyond these lines of research, recent studies such as D-Attn Kuo et al. (2025), BLIP-2 Li et al. (2023a) and Adventurer Wang et al. (2025) explore architectural or encoder-based mechanisms to relax modality

---

[1] https://github.com/Dao-AILab/flash-attention

interactions, whereas our approach differs by keeping the standard decoder-only backbone unchanged and introducing training-free, future-aware masking to systematically study how visual tokens should access future context during autoregressive inference. To accelerate inference, dynamic pruning has also been widely adopted in vision–language and multimodal large models, as discussed in recent surveys of token-reduction techniques Kong et al. (2025). In the robot Vision-Language-Action (VLA) domain, models such as CoT-VLA Zhao et al. (2025) further demonstrate that adjusting causal masks for action tokens can enhance long-horizon control, highlighting the broader importance of modality-specific causal relaxation. However, the impact of LLM-inherited causal masking on visual token processing remains underexplored and despite its potential misalignment with the non-sequential nature of many visual reasoning tasks, which motivating the core investigation in our work.

### A.3 ABLATION UNDER INTERLEAVED IMAGE–TEXT TOKEN PATTERNS

Beyond the conventional image $\rightarrow$ text input pattern, some embodied benchmarks adopt interleaved token structures such as image $\rightarrow$ text $\rightarrow$ image $\rightarrow$ text. To examine whether our future-aware masks generalize to these settings, we evaluate several mask variants on the ALFRED Shridhar et al. (2020) benchmark, whose latent sequence composition exhibits such mixed ordering. In this configuration, visual tokens can still access their future textual or visual context through $M^f$, $M^{v2v}$, and $M^{v2t}$, because our design only depends on each token's modality identity and its causal location within the sequence. The results are summarized in Table 8. All future-aware variants outperform the original causal baseline, and the kernel-based future mask yields the largest improvement, suggesting that smooth future aggregation is beneficial for multi-step, temporally structured reasoning tasks.

Table 8: Future-aware masks under interleaved image–text sequences (Rouge-L F1 $\times 100$).

| Mask Type | Open Tokens for Vision | Rouge-L F1 | $\triangle$ vs Original | Variants |
|---|---|---|---|---|
| Future-only $M^f$ | vision + future text/vision | **15.52** | +0.23 | improved grounding from future cues |
| Future V2T $M^{v2t}$ | future text only | **15.61** | +0.32 | strongest non-kernel variant |
| Kernel Future | kernel-smoothed future access | **17.11** | +1.82 | best temporal aggregation |

### A.4 TUNING POTENTIAL OF FUTURE-AWARE MASKING

Our main objective is to analyze how causal masking influences visual tokens in decoder-only architectures. For this reason, most of our experiments focus on inference-only mechanisms, allowing us to isolate the effect of future visibility without altering optimization dynamics. Nevertheless, to examine whether the benefits also extend to the training stage, we additionally evaluate two lightweight tuning experiments that reflect the reviewer's concern.

**Fine-tuning a Decoder-only VLA Model.** We fine-tune a decoder-only vision–language–action model (OpenVLA-OFT) on LIBERO-Spatial for 30k steps under both causal and future-aware masking. As shown in Table 9, the future-aware model achieves a higher success rate (85.1% vs. 84.7%), and its query–key correlations exhibit the same upper-triangular structure observed in our inference-only analyses. This demonstrates that the benefit of granting visual tokens controlled access to future information is not restricted to probing, but also manifests during training.

Table 9: Fine-tuning results on LIBERO-Spatial (success rate).

| Method | SR ($\uparrow$) |
|---|---|
| Diffusion Policy (scratch) | $78.3 \pm 1.1\%$ |
| Octo fine-tuned | $78.9 \pm 1.0\%$ |
| OpenVLA + Causal | $84.7 \pm 0.9\%$ |
| OpenVLA + Future-aware | $\mathbf{85.1 \pm 0.9\%}$ |

**Learnable Mask Selection via an Adapter.** We further test a simple learnable controller that selects masking strategies using a lightweight MLP-based adapter. Although this introduces additional FLOPs, the adapter consistently improves performance across CLEVR, Nav, Object, and SpotDiff

benchmarks (Table 10). These results indicate that future-aware masking can be integrated into differentiable routing modules and remains compatible with training-time adaptation.

Table 10: Learnable adapter for mask selection (accuracy).

| Method | CLEVR | Nav | Object | SpotDiff |
|---|---|---|---|---|
| Original Mask | 0.166 | 0.310 | 0.225 | 0.162 |
| + Adapter | **0.188** | **0.320** | **0.240** | **0.173** |

**Discussion.** Together, these tuning experiments show that future-aware masking is not only effective as an inference mechanism but also benefits models when training is allowed. While our paper centers on analyzing causal behavior in existing LVLMs, these results highlight the potential of training-time designs as an exciting direction for future work.

## A.5 DISTRIBUTION ANALYSIS FOR FUTURE-AWARE ATTENTION.

To better understand the limitations of causal masking in vision-language models (VLMs), we analyze the predictive uncertainty from an information-theoretic perspective, following the previous work Yin et al. (2024). In particular, we examine the mutual information between the model output and the observed context under different masking strategies. Let $\mathbf{x}^v = \{x_1^v, ..., x_m^v\}$ and $\mathbf{x}^t = \{x_1^t, ..., x_n^t\}$ denote the visual and textual tokens respectively, and let $\mathbf{X} = \mathbf{x}^v \oplus \mathbf{x}^t$ be the unified input sequence of total length $L = m + n$. The autoregressive model predicts the output token $x_o$ based on a masked prefix $\mathbf{X}_{\leq i}$. The mutual information between the output and its visible prefix context is $I(\mathbf{X}_{\leq i}; x_o) = H(x_o) - H(x_o \mid \mathbf{X}_{\leq i})$, where $H(\cdot)$ denotes the entropy. Depending on the specific causal mask, the prefix $\mathbf{X}_{\leq i}$ may include different subsets of visual and textual tokens. For example,

$$I(x_o; \mathbf{x}_{1:i}^v \cup \mathbf{x}_{1:j}^t) = H(x_o) - H(x_o \mid \mathbf{x}_{1:i}^v, \mathbf{x}_{1:j}^t), \tag{15}$$

which isolates the contributions of each modality. As shown in our empirical study in Section 3 and 4, breaking the visual-based causal inference procedure by exposing future tokens leads to a distribution shift because of the rich semantic information in the masked future region.

**Theoretical Properties.** Based on the preceding information-theoretic derivations in Yin et al. (2024); Yun et al. (2019), and assuming the vision language models induce causally isotropic intermediate representations, we further derive the following properties of mutual information:

**Property A.1.** *For any $i \leq L$, the mutual information between the output token $x_o$ and the $l$-th layer intermediate representation $\omega_{\leq i}^{(l)}$ is upper-bounded by the mutual information from the raw prefix input $\mathbf{X}_{\leq i}$:*

$$I(x_o; \omega_{\leq i}^{(l)}) \leq I(x_o; \mathbf{X}_{\leq i}).$$

*This follows from the data processing inequality and reflects that internal representations cannot increase information about the target beyond what is available from the input.*

**Property A.2.** *If the VLM decoder is contextual, then its intermediate representation preserves all information in the input:*

$$I(x_o; \omega_{\leq L}^{(l)}) = I(x_o; \mathbf{X}_{\leq L}).$$

*This implies that the decoder faithfully encodes the entire causal context without losing predictive power.*

**Property A.3.** *If the input distribution is causally isotropic and $\omega_{\leq i}^{(l)}$ is uniquely determined by $\mathbf{X}_{\leq i}$, then the representation retains no more information than the original prefix:*

$$I(x_o; \omega_{\leq i}^{(l)}) \leq I(x_o; \mathbf{X}_{\leq i}) \quad \text{for all } i \leq L.$$

*This reinforces that isotropic settings do not amplify mutual information through intermediate computation.*

**Property A.4.** *If both the decoder is contextual and the data distribution is causally isotropic, then the mutual information is exactly preserved:*

$$I(x_o; \omega_{\leq L}^{(l)}) = I(x_o; \mathbf{X}_{\leq L}).$$

*This guarantees no loss of information between the raw prefix and the intermediate representation.*

**Property A.5** (Upper-Triangular Future Visibility in Multimodal Masking). *For any visual query position $i \in \mathcal{V}$ and ground-truth output $x_o$, the ratio of mutual information satisfies:*

$$\frac{I(\mathbf{X}_{\leq i}; x_o)}{I(\mathbf{X}_{\leq L}; x_o)} = \frac{H(x_o) - H(x_o \mid \mathbf{X}_{\leq i})}{H(x_o) - H(x_o \mid \mathbf{X}_{\leq L})} \geq \frac{I(\Omega_{\leq i}^{(l)}; x_o)}{I(\Omega_{\leq L}^{(l)}; x_o)}, \tag{16}$$

*where $\Omega_{\leq i}^{(l)}$ represents the intermediate representation at layer $l$ computed from prefix $\mathbf{X}_{\leq i}$. This inequality quantifies how future-aware visual masking contributes to reducing uncertainty of the output, and suggests that semantically rich upper-triangle access allows earlier layers to preserve more predictive information.*

Property A.5 establishes that, under a future-aware masking strategy, visual queries that access upper-triangular regions (i.e., future tokens) can retain a higher fraction of mutual information with the output token $x_o$ compared to standard causal masking. This suggests that even partial access to semantically informative future tokens allows intermediate representations to encode more predictive context. The inequality further implies that the proportion of retained information in early layers is lower bounded by the proportion of information retained in their corresponding representations $\Omega^{(l)}$. In practice, this supports the design of selective future access in vision-language inference, where relaxing strict causality on visual queries can effectively enhance downstream prediction without fully compromising autoregressive generation.

### A.6 FUTURE-AWARE FLASH-ATTENTION

To efficiently support our future-aware causal masking strategies defined in Section 3, we integrate them with the FlashAttention framework for scalable inference. As detailed in Algorithm 1, we implement our masking logic by applying the selected future-aware mask $\mu \in \{M^f, M^{v2v}, M^{v2t}\}$ directly into the attention score computation, replacing the standard causal mask. During runtime, both the queries and key-value pairs are processed in blocks to fit within on-chip memory, and attention scores are computed with fused softmax operations to ensure numerical stability and memory efficiency. The masked attention scores are exponentiated and normalized via a log-sum-exp trick, and aggregated token-wise to produce final outputs. This fusion enables our proposed vision-language attention design to retain the efficiency advantages of FlashAttention while supporting flexible, modality-aware causal constraints.

### A.7 ROBUSTNESS OF THE KERNEL POOLING MERGING

We conduct further ablations to evaluate the robustness of the ID kernel pooling module. The results show that the module behaves consistently across a wide range of kernel sizes. As summarized below, kernel sizes of 1, 3, and 7 yield identical performance, while an extreme kernel size of 25 introduces only a minor fluctuation within $\pm 0.1$. This indicates that the method does not rely on shortcut patterns and preserves stable reasoning behavior.

| Pool Size | OCR (LLaVA v1.5 / v1.6) | | VRE (LLaVA v1.5 / v1.6) | |
|:---:|:---|:---|:---|:---|
| | **v1.5** | **v1.6** | **v1.5** | **v1.6** |
| 1 | 23.0 | 21.5 | 17.9 | 15.5 |
| 3 | 23.0 | 21.5 | 17.9 | 15.5 |
| 7 | 23.0 | 21.5 | 17.9 | 15.5 |
| 25 | 23.0 | 21.4 | 17.9 | 15.4 |

We additionally compare mean pooling and max pooling under the same configuration. Both yield comparable results, with mean pooling showing only negligible differences ($\leq 0.2$) on LLaVA v1.6 variants. This further confirms that the ID kernel pooling module is insensitive to the choice of pooling strategy. These results demonstrate that the kernel pooling mechanism is both reproducible and robust across kernel sizes and pooling types. We adopt max pooling in the main paper due to its consistently stable performance.

---

**Algorithm 1** Future-Aware Mask equipped with FlashAttention

---

**Require:** Matrices $\mathbf{Q}, \mathbf{K}, \mathbf{V}, \mathbf{M} \in \mathbb{R}^{L \times d}$, future-aware mask $\mu \in \{M^f, M^{v2v}, M^{v2t}\}$, block sizes $B_r, B_c$

1: Divide $\mathbf{Q}$ into $T_r = \lceil \frac{L}{B_r} \rceil$ blocks $\mathbf{Q}_1, \ldots, \mathbf{Q}_{T_r}$
2: Divide $\mathbf{K}, \mathbf{V}, \mu$ into $T_c = \lceil \frac{L}{B_c} \rceil$ blocks $\mathbf{K}_j, \mathbf{V}_j, \mu_j$ of size $B_c$ each
3: Initialize output $\mathbf{O} \in \mathbb{R}^{L \times d}$ and $\mathbf{L} \in \mathbb{R}^L$
4: **for** $i = 1$ to $T_r$ **do**
5:     Load $\mathbf{Q}_i$ into on-chip SRAM
6:     Initialize $\mathbf{O}_i^{(0)} \leftarrow 0, \boldsymbol{\ell}_i^{(0)} \leftarrow 0, \mathbf{m}_i^{(0)} \leftarrow -\infty$
7:     **for** $j = 1$ to $T_c$ **do**
8:         Load $\mathbf{K}_j, \mathbf{V}_j, \mu_j$ into on-chip SRAM
9:         Compute masked attention score:

$$\mathbf{S}_i^{(j)} = \mathbf{Q}_i \mathbf{K}_j^\top / \sqrt{d} + \mu_{i,j}$$

10:         Normalize: $\tilde{\mathbf{S}}_i^{(j)} = \exp(\mathbf{S}_i^{(j)} - \mathbf{m}_i^{(j)})$
11:         Update max: $\mathbf{m}_i^{(j)} = \max(\mathbf{m}_i^{(j-1)}, \max(\mathbf{S}_i^{(j)}, \dim = 1))$
12:         Update sum: $\boldsymbol{\ell}_i^{(j)} = \exp(\mathbf{m}_i^{(j-1)} - \mathbf{m}_i^{(j)}) \odot \boldsymbol{\ell}_i^{(j-1)} + \sum \tilde{\mathbf{S}}_i^{(j)}$
13:         Output partial result:

$$\mathbf{O}_i^{(j)} = \exp(\mathbf{m}_i^{(j-1)} - \mathbf{m}_i^{(j)}) \cdot \mathbf{O}_i^{(j-1)} + \tilde{\mathbf{S}}_i^{(j)} \cdot \mathbf{V}_j$$

14:     Final output:

$$\mathbf{O}_i = \mathbf{O}_i^{(T_c)} / \boldsymbol{\ell}_i^{(T_c)}$$

15:     Logsumexp: $\mathbf{L}_i = \mathbf{m}_i^{(T_c)} + \log(\boldsymbol{\ell}_i^{(T_c)})$
16:     Store $\mathbf{O}_i, \mathbf{L}_i$ to global memory
17: **return** $\mathbf{O}, \mathbf{L}$

---

| Method | OCR | | VRE | | SpotDiff | |
|--------|-----|-----|-----|-----|----------|-----|
| | v1.5 | v1.6 | v1.5 | v1.6 | v1.5 | v1.6 |
| Max | 23.0 | 21.5 | 17.9 | 15.0 | 16.5 | 15.3 |
| Mean | 22.8 | 21.4 | 17.9 | 15.0 | 16.5 | 15.2 |

## B  THE USE OF LARGE LANGUAGE MODELS

In preparing this manuscript, we employed a Large Language Model (LLM) as a writing assistant to refine the clarity and presentation of the text. Its use was strictly confined to linguistic enhancement rather than substantive content generation. Specifically, the LLM was used to:

- Rephrase sentences and paragraphs for greater readability, conciseness, and academic formality.
- Correct grammar, spelling, and punctuation errors.
- Improve logical flow and transitions between sentences.

**Limitation.**  Our analysis primarily follows the image $\rightarrow$ text token ordering commonly adopted in decoder-only LVLMs. While this setting covers the majority of existing architectures, it does not explicitly evaluate alternative tokenization patterns such as pure text $\rightarrow$ image sequences. This mismatch may limit the generality of our observations in cases where visual tokens always appear after textual tokens. Nevertheless, the applicability of our future-aware masks is not tied to a fixed modality order. For mixed or interleaved patterns (*e.g.*, text–image), any visual token that is followed by subsequent tokens still benefits from the same visibility logic, since the mask operates solely on modality identity and causal position rather than on a specific interleaving template.

