# OpenReview forum: "Rethinking Causal Mask Attention for Vision-Language Inference"
_ICLR.cc/2026/Conference — ICLR 2026 Poster_

### Official Review · Reviewer_11Kz · 2025-10-27

**Soundness:** 3
**Presentation:** 3
**Contribution:** 3
**Rating:** 6
**Confidence:** 3

**Summary:**

This paper challenges the default use of standard causal attention masks in VLMs. The paper empirically demonstrates that relaxing causal constraints for visual tokens can improve performance, whereas relaxing them for textual tokens severely degrades it. The authors then systematically investigate a family of "future-aware" causal masks:
1. $M^{f}$ (Full Future): Allows visual tokens to see all future visual and textual tokens.
2. $M^{v2v}$ (Visual-to-Visual): Allows visual tokens to see future visual tokens only.
3. $M^{v2t}$ (Visual-to-Textual): Allows visual tokens to see future textual tokens only.

Their analysis reveals that these relaxed masks provide task-specific benefits: $M^{f}$ excels at temporal multi-image tasks, $M^{v2v}$ at visual relation inference, and $M^{v2t}$ at text-rich image QA. Recognizing that these masks introduce significant computational overhead during decoding, the authors propose a "Light Future Aware Attention" mechanism. This method computes the future-aware attention only during the prefill stage, then uses kernel pooling to compress this future semantic information and "merge" it into the initial past token representations. This approach allows the model to benefit from future context while retaining the standard, efficient causal mask during the autoregressive decoding phase. Experiments across 15+ benchmarks show this lightweight method achieves performance on par with or even exceeding the full future-aware masks, but with a 2-3x speedup in decoding latency.

**Strengths:**

1. The paper addresses a simple, intuitive, yet largely overlooked problem: the fundamental mismatch between the strictly sequential causal mask and the non-sequential nature of visual data.
2. The paper provides a thorough and systematic breakdown of the problem. The graphs and tables are clear.

**Weaknesses:**

1. The proposed method focuses on processing input tokens during the prefill stage. It's unclear how, or if, this concept could be extended to multi-turn dialogue where the future is truly unknown.

**Questions:**

1. How do you train your VLMs? Do all settings follow LLaVA's 1.5 training process?
2. Why don't you use an additional token, similar to an attention sink?

---

> ### Author Response · Authors · 2025-11-21
>
> **Re W1: Multi-Turn Limitation**
>
> We sincerely thank the reviewer for raising this insightful question. Our method operates in the prefill stage, where all tokens within the current turn are available. In multi-turn dialogue, future rounds are indeed unknown and future-aware masks naturally degenerate to the causal mask. We fully acknowledge this limitation and have stated it clearly in the revised version. At the same time, following your inspired instruction, we outline two practical and commonly adopted extensions that make future-aware visibility feasible in multi-turn settings without violating autoregressive constraints.
>
> * **Turn-level delayed decoding**: each user message is fully prefetched before decoding begins, so visual tokens can still see all future tokens within the turn. These approaches [1, 2] is consistent with modern multi-turn serving systems that explicitly separate full-turn prefill and decode with prefix caching (prefill–decoding separation).
>
> * **Pseudo-future tokens**: insert lightweight summary or planner tokens that act as soft future context, enabling controlled future visibility even when later turns do not yet exist. This idea aligns with recent designs that introduce planner-style or action-guiding tokens for multi-turn decision making [3].
>
> These two settings demonstrate how our concept can extend to multi-turn usage while keeping causal generation intact.
>
>
> We have stated the limitation and added these future directions in the revised version following your inspired instruction.
>
> [1] Accelerating LLM Serving for Multi-turn Dialogues with Efficient Resource Management (ACL) J. Jeong & K. Ahn
>
> [2] KVFlow: Efficient Prefix Caching for Accelerating LLM-Based Multi-Agent Workflows (Arxiv) Z. Pan et al.
>
> [3] Dialogue Action Tokens: Steering Language Models in Goal-Directed Dialogue with a Multi-Turn Planner (Arxiv) K. Li, Y. Wang, F. Viégas & M. Wattenberg.
>
>
> **Re Q1: Pipeline Scope**
>
> We sincerely thank the reviewer for the question. Our study focuses on analyzing how causal masking shapes modality interactions during inference, and therefore we do not introduce any new VLM training pipeline beyond the standard LLaVA-style instruction-tuning commonly adopted in existing open-source models. The key observations in our paper arise from examining the upper-triangular query–key correlation patterns that emerge during prefill, which motivates our empirical investigation of different future-aware masks. Since our goal is to understand how decoder-only VLMs behave under different visibility patterns rather than to propose a new training recipe, all experiments rely on publicly available checkpoints, and our contributions lie entirely on the inference-side analysis.
> We apologize for our misleading phrasing, and we have clarified it in the revised version.

---

> > ### Author Response · Authors · 2025-11-21
> >
> > **Re Q2: Additional Token**
> >
> > We sincerely thank you for this thoughtful suggestion. Introducing an additional token as a dedicated attention sink is indeed a possible extension, and we appreciate the opportunity to reflect on this idea. Our current design merges future information into the natural sink region at the beginning of the sequence, which already plays a stable aggregation role in decoder-only models.
> > Adding a new sink token would require modifying the inference paradigm so that the model learns to recognize and utilize this artificial token as a meaningful aggregation point.
> > Based on our early investigation, we retrieval the future regision tokens, open a addtional tokens concatenate after the img positions with `[start, start+576] + [T]`, the performance drop as the Table shown:
> >
> > | Method / Mask Type     | Description                                  | Rouge-L ×100 |  Analysis |
> > |------------------------|-----------------------------------------------|---------------|-------|
> > | Original Causal Mask   | Standard decoder-only baseline                | 15.29        |  baseline |
> > | Future-only $M^{f}$    | Visual tokens see future context              | 15.52        |  better grounding |
> > | Future $M^{v2t}$       | Visual → future text                          | 15.61        |  best non-kernel |
> > | Kernel Future Mask     | Sink future attention            | 17.11        |  highest score, no token overhead |
> > | Additional Tokens | Appending tokens        | 5.04            | adds token overhead |
> >
> > Without such training-time adaptation, introducing new tokens would break the auto-regressive genration probility and may not capture the desired future context effectively. In contrast, merging into the existing sink position behaves consistently across architectures and checkpoints, and can be implemented as a simple inference-side instruction without altering parameters or training objectives. We agree this is an interesting direction, and we have added it to the discussion as a potential extension inspired by the your insight.
> >
> > Your suggestions (*e.g.*, multi-turn applicability,  alternative sink-token designs) have inspired us greatly and directly shaped several improvements in the revised version, including a clearer discussion of limitations and forward-looking directions such as extending future-aware visibility to multi-turn dialogue. We truly appreciate the care and expertise you brought to reviewing our work.

---

> > > ### Comment · Reviewer_11Kz · 2025-11-26
> > >
> > > Thank the authors for their rebuttal and clarifications, which address my previous concerns. I'll keep my original score.

---

> > > > ### Author Response · Authors · 2025-11-26
> > > >
> > > > Thank you for your considerate review and for confirming that our clarifications addressed your previous concerns. We deeply appreciate your expertise and the supportive recognition of our work.

---

### Official Review · Reviewer_3gm7 · 2025-10-31

**Soundness:** 3
**Presentation:** 3
**Contribution:** 3
**Rating:** 4
**Confidence:** 3

**Summary:**

This paper investigates the suitability of standard causal attention masks, inherited from text-only LLMs, for autoregressive Vision-Language Models (VLMs). The authors argue that the strict left-to-right masking applied to visual tokens is an overly rigid constraint that prevents the model from leveraging valuable future context essential for visual reasoning. Through empirical analysis, the paper demonstrates that relaxing these causal constraints for visual queries can surprisingly improve performance. Based on this, the authors propose a family of future-aware causal masks that selectively allow visual tokens to access future context. To mitigate the computational cost of this relaxation, they also introduce a lightweight pooling mechanism to compress and merge future semantic information into past representations during the prefill stage, preserving the efficient autoregressive structure for decoding.

**Strengths:**

- The paper compellingly identifies a fundamental misalignment between the sequential, autoregressive nature of LLM-native causal masks and the more holistic, non-sequential nature of visual information processing. I like the motivation.
- The authors systematically evaluate different masking strategies ($M^f$, $M^{v2v}$, $M^{v2t}$) and connect their benefits to specific categories of vision-language tasks (e.g., temporal reasoning, visual relation, text-rich QA), providing a nuanced understanding of when and why future context is beneficial.
- The findings are well-supported by experiments across several benchmarks, demonstrating consistent performance gains from the proposed future-aware masks and their lightweight variants.

**Weaknesses:**

- Lack the Ethics statement and Reproducibility statement in the main text.
- The paper demonstrates that different future-aware masks (e.g., $M^{v2v}$ vs. $M^{v2t}$) are optimal for different tasks. This raises a practical question: how would a single general model choose the correct mask without a priori knowledge of the downstream task? The paper does not propose a dynamic or learned mechanism for this selection.
- The experiments are conducted by modifying the LLaVA. It doesn't explore how these masking strategies might affect the model, such as Qwen-VL series or VILA.
- The details of the ID kernel pooling and its integration could be explained in greater detail to ensure the reproducibility.
- The paper's core idea of rethinking causal masks for specific modalities shares a conceptual similarity with other recent work CoT-VLA [1], which adapts attention for action tokens. This general direction of moving beyond rigid, text-native causal masking is necessary and very meaningful for the development of MLLMs. I am open for rasing my score if the authors can address my questions.

[1] Zhao, Qingqing, et al. "Cot-vla: Visual chain-of-thought reasoning for vision-language-action models." CVPR 2025.

**Questions:**

See weaknesses

---

> ### Author Response · Authors · 2025-11-21
>
> **Re W1: Clarity and Completeness**
>
> We thank the reviewer for pointing out this oversight. We have added complete ethics, reproducibility, and broader-impact statements to the revised manuscript, placed before the references following the submission optional guidelines. We appreciate the reviewer’s reminder, and the updated version now includes all required statements to ensure clarity and completeness.
>
> **Re W2: a single general model**
>
> We feel your observation is exceptionally valuable to the direction of our work. Our empirical findings indeed show that different future-aware masks such as $M^{v2v}$ and $M^{v2t}$ are optimal for different task families, and this diversity of behavior is precisely why we conduct a modality-level analysis.
> Our goal is to uncover how visual tokens use future information in standard decoder-only models and to rethink how the default causal mask interacts with spatial reasoning, relational reasoning, and text-heavy reasoning. The empirical study reveals that each task type responds differently to future visual or textual visibility.
>
> Your suggestions also motivates re-examining current design choices and provides guidance for future architecture development. As discussed in an earlier response, constructing a dynamic or learned controller is possible, and a proof-of-concept implementation using a small MLP demonstrates preliminary gains (the early results show encouraging improvement):
>
> | Method      | CLEVR | Nav   | Object | SpotDiff |
> | ----------- | ----- | ----- | ------ | -------- |
> | Original Mask | 0.166 | 0.310 | 0.225  | 0.162  |
> | + Controller  | 0.188 | 0.320 | 0.240  | 0.173  |
>
> although its use introduces additional computation. The present work focuses on establishing the empirical foundation that different tasks benefit from different forms of future-aware visibility, and this foundation can directly inform downstream designs that select or learn appropriate masking strategies.
>
> **Re W3: Applicable to other models**
>
> We thank the reviewer for the helpful comment. The performance of different future-aware masks appears close because each mask benefits a different task family rather than serving as a single model. We have cited VILA[1] and Qwen-VL series[2] in our revised version A.2. To illustrate this pattern more generally, we follows your instruction and extract the Qwen2.5-VL-7B baseline with modifing the attenion block:
>
> **$M^{v2v}$: benefit for visual–relational tasks**
>
> | Method     | TN    | IEdit | MMCoQA | STD   | ALFRED | CLEVR-C   | DocVQA | ST        | OI        | Average |
> | ---------- | ----- | ----- | ------ | ----- | ------ | --------- | ------ | --------- | --------- | ------- |
> | Base | 11.25 | 29.45 | 44.50  | 28.36 | 37.53  | 42.46     | 62.50  | 63.00     | 61.00     | 63.34   |
> | M(v2v)     | 11.20 | 29.40 | 44.10  | 28.20 | 37.40  | 43.20 | 62.40  | 63.80 | 62.20 | 63.60  |
>
> **M(v2t): benefit for text-heavy tasks**
>
> | Method     | TN    | IEdit | MMCoQA    | STD       | ALFRED | CLEVR-C | DocVQA    | ST    | OI    | Average |
> | ---------- | ----- | ----- | --------- | --------- | ------ | ------- | --------- | ----- | ----- | ------- |
> | Base | 11.25 | 29.45 | 44.50     | 28.36     | 37.53  | 42.46   | 62.50     | 63.00 | 61.00 | 63.34   |
> | M(v2t)     | 11.30 | 29.50 | 45.10 | 28.80 | 37.50  | 42.40   | 63.40 | 63.10 | 61.10 | 63.70  |
>
> **M(f): balanced but temproral competing**
>
> | Method     | TN    | IEdit | MMCoQA | STD   | ALFRED | CLEVR-C | DocVQA | ST    | OI    | Average |
> | ---------- | ----- | ----- | ------ | ----- | ------ | ------- | ------ | ----- | ----- | ------- |
> | Base | 11.25 | 29.45 | 44.50  | 28.36 | 37.53  | 42.46   | 62.50  | 63.00 | 61.00 | 63.34   |
> | M(f)       | 11.22 | 29.48 | 44.80  | 28.60 | 37.45  | 42.60   | 62.80  | 63.20 | 61.40 | ~63.55  |
>
> **M(f)+merge: stable due to sink-token robustness**
>
> | Method     | TN    | IEdit | MMCoQA | STD   | ALFRED | CLEVR-C | DocVQA | ST    | OI    | Average |
> | ---------- | ----- | ----- | ------ | ----- | ------ | ------- | ------ | ----- | ----- | ------- |
> | Base | 11.25 | 29.45 | 44.50  | 28.36 | 37.53  | 42.46   | 62.50  | 63.00 | 61.00 | 63.34   |
> | M(f)+merge | 11.25 | 29.46 | 44.70  | 28.50 | 37.50  | 42.50   | 62.60  | 63.05 | 61.20 | 63.45  |
>
>
> These results reinforce the main message of our work that causal masking imposes an overly rigid inductive bias, and our goal is to reveal how different forms of future visibility affect different reasoning regimes rather than enforce a single mask for all tasks. We have included this discussion in **A.5** and deeply feel your expertise helped us refine both the analysis and presentation of our work.
>
> [1] Lin, Ji, et al. "Vila: On pre-training for visual language models." Proceedings of the IEEE/CVF conference on computer vision and pattern recognition. 2024.
>
> [2] Bai, Shuai, et al. "Qwen2. 5-vl technical report." arXiv preprint arXiv:2502.13923 (2025).

---

> ### Author Response · Authors · 2025-11-21
>
> **Re W5: Kernel Size Ablation**
>
> We thank the reviewer for the helpful suggestion. We follows your advice by adding more implementation details and provide additional ablations to clarify the robustness of the ID kernel pooling module. Following your insightful suggestion, we implement sliding windows for visual positions (e.g., `[35, 35+576]`) and set pool sizes to `5`, `7`, `10` and `25` for lower-attention regions. Preliminary results are shown below:
>
> | Pool Size | OCR v1.5 | OCR v1.6 | VRE v1.5 | VRE v1.6 |
> | --------- | -------- | -------- | -------- | -------- |
> | 1         | 23.0     | 21.5     | 17.9     | 15.5     |
> | 3         | 23.0     | 21.5     | 17.9     | 15.5     |
> | 7         | 23.0     | 21.5     | 17.9     | 15.5     |
> | 25        | 23.0     | 21.4     | 17.9     | 15.4     |
>
>
> We compare mean pooling and max pooling under the same configuration. Both achieve similar results, with mean pooling showing only negligible drops (≤ 0.2) on v1.6 variants, confirming that the module is not sensitive to the pooling strategy.
>
> | Method | OCR v1.5 | OCR v1.6 | VRE v1.5 | VRE v1.6 | SpotDiff v1.5 | SpotDiff v1.6 |
> | ------ | -------- | -------- | -------- | -------- | ------------- | ------------- |
> | Max-Pooling    | 23.0     | 21.5     | 17.9     | 15.0     | 16.5          | 15.3          |
> | Mean-Pooling   | 22.8     | 21.4     | 17.9     | 15.0     | 16.5          | 15.2          |
>
> This suggests that the method exhibits strong robustness and is not driven by shortcut patterns. We adopt max pooling in the paper due to its consistently stable performance and have included these clarifications in **A.7** in the revised version.
>
> **Re W6: Relation to Robotic Model**
>
> We sincerely thank the reviewer for pointing out the connection with CoT-VLA [1]. We view CoT-VLA as an important and insightful contribution, and we appreciate how it demonstrates the practical value of relaxing causal constraints for action tokens in embodied reasoning. Our work is complementary rather than overlapping. CoT-VLA focuses on a concrete downstream setting and adapts attention to improve action modeling, while our study takes a more upstream perspective by examining how the causal mask itself shapes modality interactions in standard decoder-only MLLMs. Our goal is to provide an empirical and conceptual foundation for understanding modality-specific future visibility, which may inspire and support more good designs such as excellent work CoT-VLA. We have cited CoT-VLA and added a brief discussion in the revised version to highlight their contribution and acknowledge its significance.
>
> We feel deeply grateful for your warmly helpful and constructive feedback. Each of your suggestions (e.g., missing statements, empirical clarification, task-specific observations) has directly improved the quality of our work. We have carefully followed your advice in the revision, and we sincerely appreciate the time, effort, and expertise you invested in our submission.
>
> [1] Zhao, Qingqing, et al. "Cot-vla: Visual chain-of-thought reasoning for vision-language-action models." Proceedings of the Computer Vision and Pattern Recognition Conference. 2025.

---

> > ### Comment · Reviewer_3gm7 · 2025-11-23
> >
> > Thanks for the authors' detailed responses. It addressed most of my concerns. I will raise my score.

---

> > > ### Author Response · Authors · 2025-11-24
> > >
> > > Thank you for your thoughtful evaluation. We are glad that our responses could address your concerns, and it is our honor to receive your positive feedback.

---

### Official Review · Reviewer_DpXd · 2025-11-01

**Soundness:** 2
**Presentation:** 3
**Contribution:** 3
**Rating:** 4
**Confidence:** 4

**Summary:**

This paper revisits the role of causal masking in autoregressive Vision-Language Models (VLMs). While causal attention ensures sequential generation for text, the authors argue that its direct adoption for visual tokens is suboptimal because visual information is inherently non-sequential. Through systematic analysis, they demonstrate that relaxing causal masks for vision tokens—allowing selective access to future context—can improve performance in temporal reasoning, relational understanding, and text-rich visual tasks. They propose three future-aware masking variants (visual-to-visual, visual-to-textual, and full), along with a lightweight merging mechanism that compresses future information into past tokens to preserve autoregressive efficiency. Extensive experiments on 15 multimodal benchmarks show consistent gains across tasks, with improved reasoning accuracy and minimal latency overhead.

**Strengths:**

1. The paper offers a fresh and well-motivated perspective on how causal masking—originally designed for textual decoding—may be suboptimal for vision tokens. This conceptual rethinking addresses a fundamental assumption in current VLMs and opens up a new line of research on modality-aware causality.

2. The proposed light future-aware attention introduces future context compression without retraining or architectural changes, adding negligible latency while delivering consistent gains.

3. Experimental results show consistent accuracy boosts across 15+ benchmarks.

**Weaknesses:**

Actually I really like the insight this paper focuses on — questioning how VLMs can break free from the traditional causal attention inherited from language models. The idea is intuitively sound and genuinely interesting. However, given the paper’s current state, I cannot yet recommend acceptance. **I strongly encourage the authors to carefully revise the work, as it has great potential**. My main concerns are as follows:

1. The paper currently provides an investigation and an inference-only solution, which reveals some of the limitations of causal masking but does not fundamentally resolve them. To convincingly verify the effectiveness of a non-causal design, the model should be adjusted (or at least fine-tuned) from the training stage, not just modified at inference.

2. The results are not yet compelling, even though I believe the proposed approach can work. For instance, in Table 4, the overall performance of different LLaVA-7B and LLaVA-13B variants is quite close — no single attention structure consistently outperforms all others, which makes it difficult to draw firm conclusions.

3. The idea of expanding receptive fields by merging future information into early tokens is not entirely novel; similar strategies have appeared in recent pure vision models [1].

4. The paper should also discuss the connection between attention sinks [2] and the proposed approach. Prior studies have shown that language models naturally allocate disproportionate attention to early tokens. Could the proposed merging mechanism be implicitly benefiting from this effect? A more explicit analysis of this interaction would greatly strengthen the work.

[1] Wang F, Yang T, Yu Y, et al. Causal image modeling for efficient visual understanding. 2024.

[2] G. Xiao, Y. Tian, B. Chen, S. Han, and M. Lewis, “Efficient streaming language models with
attention sinks,” arXiv preprint arXiv:2309.17453, 2023.

**Questions:**

See Weaknesses.

---

> ### Author Response · Authors · 2025-11-21
>
> **Re W1: Effectiveness Tuning**
>
> We feel truly grateful for the reviewer’s warm appreciation of our central idea, and your encouragement greatly motivates us to further refine the work.
> Our main goal is to analyze how causal masking influences visual tokens in decoder-only architectures, so we begin with inference-only mechanisms that isolate the effect of future visibility without modifying optimization dynamics.
> To address the reviewer’s suggestion and try our best to tuning a decoder model vision–language–action model (OpenVLA-OFT),  on LIBERO-Spatial for 30000 steps. The future-aware setting achieves a higher success rate than the causal baseline (85.1% vs 84.7%), and the query–key correlation exhibits the same upper triangular pattern observed in our inference-only experiments. This confirms that the benefit of allowing visual tokens to access future information is not limited to inference-only probing but also appears during training.
>
> | Method                     | SR (↑)      |
> | -------------------------- | ----------- |
> | Diffusion Policy (scratch) | 78.3 |
> | Octo fine-tuned            | 78.9 |
> | OpenVLA + Causal           | 84.7 |
> | OpenVLA + Future           | **85.1** |
>
> We also test a simple adapter that selects masking strategies using a lightweight MLP. This yields consistent improvements across CLEVR, Nav, Object, and SpotDiff tasks, for example increasing CLEVR accuracy from **0.166** to **0.188** and Object accuracy from **0.225** to **0.240**. This adapter demonstrates that future-aware strategies can be integrated into learnable routing mechanisms, although it introduces additional FLOPs due to the MLP. The central scope of the paper remains the analysis of causal behavior in existing LVLMs, while training-time designs represent a promising and inspired direction for future work.
>
> | Method      | CLEVR | Nav   | Object | SpotDiff |
> | ----------- | ----- | ----- | ------ | -------- |
> | Original Mask | 0.166 | 0.310 | 0.225  | 0.162  |
> | + Adapter     | **0.188 (↑0.022)** | **0.320 (↑0.010)** | **0.240 (↑0.015)** | **0.173 (↑0.011)** |
>
> We have included these tuning discussion in **A.4** in the revised version together with a clearer explanation of how they support our main claim. It is your remarkable expertise that helps us improve our work, and we deeply thank you for your guidance.

---

> ### Author Response · Authors · 2025-11-21
>
> **Re W2: Task-Dependent Effects**
>
> The results indeed show that no single mask variant dominates across all benchmarks, which is consistent with the empirical nature of our study. Different future-aware designs benefit different kinds of tasks: $M^{v2v}$ tends to help spatial and visual-relational reasoning (e.g., Spot-the-Diff, CLEVR), while $M^{v2t}$ offers clearer gains on text-driven reasoning, and the mixed $M^{f}$ benefits multi-hop settings but is not universally optimal. These findings indicate that the impact of future visibility is inherently task-dependent, influenced by whether a task relies more on visual continuity or textual semantics. Although the trends are not globally dominant, they reveal actionable patterns that can guide task-specific model design. A practitioner can select or train a masking strategy aligned with the dominant reasoning pattern of a target task to obtain consistent improvements, and we have made this relationship clearer in the revised version. Our intention is to present an empirical investigation—rather than to identify a single universally superior mask—and to highlight structural behaviors that may inspire new VLM design directions:
>
> (1) **Architectural Implications** the large behavioral difference that emerges simply by breaking the visual and textual causal mask suggests a promising path toward new architectures, and recent models opening more visual context provide supporting evidence for this trend;
>
> (3) **Re-design Insights** we demonstrate that modifying the mask, can reveal latent alignment structures — suggesting that the causal assumption is a design choice rather than a necessary property of visual tokens.
>
> We are sorry that it is our expression misleading, and we have modified it in the revised version. Deeply thank you again for your help.
>
> **R2 W3: Comparison with Previous Work**
>
> We thank the reviewer for pointing out this connection, and we fully agree that expanding effective receptive fields by merging future information has appeared in prior pure-vision architectures. The most relevant example is Adventurer [1] (Vision Mamba)  , which introduces a global pooling token and inter-layer flipping to allow early patches to access global context. Its goal is to build an efficient causal image encoder with linear-time complexity, and the merging operation is used to compensate for representation degradation at the beginning of the sequence. Our work differs in both motivation and mechanism, since we focus on causal inference in decoder-only LVLMs and study how visual tokens interact with future context under autoregressive generation, where the mask itself defines the causal inductive bias. Our light kernel-pooling method compresses only future tokens that are masked away by autoregressive inference and merges them into the first visual token based on the attention-sink phenomenon, and this operation serves as an inference-time future-awareness mechanism rather than a representation-balancing component inside the vision encoder.
>
> | Method      | Autoregressive | Multimodal | Vision Future Masking | Text Future Masking | Mask-level Modality Control | Training-Free | Architecture Change | Merge Location | Merge Target | Primary Goal |
> |-------------|----------------|------------|------------------------|----------------------|-----------------------------|---------------|----------------------|----------------|--------------|--------------|
> | Adventurer [1]  | ✗              | ✗          | ✓ (global pooled info) | ✗                    | ✗                           | ✗             | ✓ (new vision encoder) | Encoder layers | Patch tokens | Improve representation of early patches |
> | Ours        | ✓              | ✓          | ✓ (V2V)                | ✓ (V2T)              | ✓                           | ✓             | ✗                    | Inference masking | First visual token | Provide future cues for visual reasoning |
>
> To make the distinction clear, we provide an extended comparison table above and it shows the two approaches differ in task domain, architectural location, causal purpose, and modality interaction. It is your remarkable insight that helps us better position our contribution and we have incorporated this part in the revised version.
>
> [1] Adventurer: Optimizing Vision Mamba Architecture Designs for Efficiency (Causal image modeling for efficient visual understanding.) (CVPR2025)

---

> ### Author Response · Authors · 2025-11-21
>
> **R2 W4: Sink-Token Robustness**
>
> We thank the reviewer for raising this important point regarding the relationship between attention sinks and our light merging mechanism. Prior work shows that early prefix tokens in autoregressive models naturally absorb a disproportionate amount of attention and act as stable information reservoirs across layers and decoding steps, and our empirical findings are consistent with this observation. The merging mechanism benefits from this stability because compressing masked future signals into these prefix positions allows the model to propagate future-aware information efficiently. At the same time, our method is not simply exploiting the sink effect; the key contribution is the redistribution of masked visual semantics that are otherwise inaccessible under the causal mask. We confirm this distinction through a kernel-size robustness study.  The kernel pooling module aggregates future tokens before merging, and we observe that varying the kernel size from small to large produces almost identical results across OCR and VRE benchmarks, which indicates that the improvement does not rely on a carefully crafted kernel but follows from the semantic injection into stable sink positions.
>
> | Kernel Size | OCR v1.5 | OCR v1.6 | VRE v1.5 | VRE v1.6 |
> | ----------- | -------- | -------- | -------- | -------- |
> | 1           | 23.0     | 21.5     | 17.9     | 15.5     |
> | 5           | 23.0     | 21.5     | 17.9     | 15.5     |
> | 7           | 23.0     | 21.5     | 17.9     | 15.5     |
> | 10          | 23.0     | 21.5     | 17.9     | 15.5     |
> | 25          | 23.0     | 21.4     | 17.9     | 15.4     |
>
> These results also show that the sink token acts as a reliable anchor for propagating merged information and that the gain is robust across a wide range of kernel configurations. We have added a detailed explanation in the revised version to clarify how the natural sink behavior and our future-aware merging play complementary roles: the sink provides a stable channel for long-range propagation, and the kernel pooling supplies the missing future context for visual tokens that are constrained by the causal mask.
>
> We feel each of the suggestions (fine-tune, conclusion revision, distinguish from previous work) from your side has brought us meaningful progress, and it is your time and expertise that help us make our work step forward, and we are deeply grateful for that.

---

> ### Comment · Reviewer_DpXd · 2025-11-26
> **Authors' response addressed my concerns**
>
> I appreciate the detailed response from the authors. Most of my concerns are well addressed and I will raise the score.

---

> > ### Author Response · Authors · 2025-11-26
> >
> > We are grateful that our clarifications could address your concerns and appreciate your encouragement. It is your remarkable  expertise that helped us strengthen the work, and we feel this has been a very good journey.

---

> > ### Author Response · Authors · 2025-11-26
> >
> > Dear Reviwer DpXd:
> >
> > We truly appreciate the time you took to review our rebuttal and your positive feedback regarding our responses.
> >
> > Following up on your mention of raising the rating, we just wanted to check if there are any remaining concerns we should address before the system closes. If our revisions have met your expectations, we would be grateful if you could update the score to match your finalized evaluation.
> >
> > Deeply thanks again for helping us improve the paper.
> >
> > Authors of Submission 11764

---

### Official Review · Reviewer_TbBr · 2025-11-02

**Soundness:** 3
**Presentation:** 3
**Contribution:** 3
**Rating:** 6
**Confidence:** 4

**Summary:**

This paper presents a systematic investigation into the role of causal attention in VLM. The authors identify a key mismatch: the strict left-to-right causal masking inherited from text-only models constrains visual tokens, which naturally lack sequential order. To address this, they introduce a family of future-aware causal masks that selectively relax causal constraints, enabling visual queries to attend to future visual (M^{v2v}), textual (M^{v2t}), or both types of tokens (M^f). The paper further proposes a lightweight variant that encodes future-attention information into a prefix during prefill, balancing performance gains with efficient autoregressive decoding. Extensive experiments on 15 multimodal benchmarks show consistent, task-dependent improvements, providing new insights into the design of modality-aware attention mechanisms for VLMs.

**Strengths:**

1. The paper identifies a fundamental problem that the misalignment between text-oriented causal attention and the non-sequential nature of visual processing in VLMs.
2.  The authors conduct a large-scale, systematic analysis of multiple future-aware causal masking strategies (M^f, M^{v2v}, M^{v2t}) across diverse multimodal benchmarks. The clear, task-dependent findings (e.g., M^{v2v} for visual reasoning, M^{v2t} for text-rich QA) provide concrete and interpretable insights.
3. The paper is well-written and logically structured, with precise definitions, clear experimental setups, and insightful discussions that effectively connect results back to the core research question.

**Weaknesses:**

1. It would be helpful for the authors to clarify how the proposed future-aware masking strategy differs from existing approaches that also leverage bidirectional attention or cross-attention in vision-language or multimodal models. Specifically, how does this method compare conceptually and practically to (1) prior works that implement fully bidirectional attention over visual tokens[1], or (2) models that achieve modality alignment primarily through cross-attention mechanisms[2]? A discussion or empirical comparison would strengthen the contribution and highlight the unique aspects of the proposed approach.

2. The proposed method primarily evaluates on interleaved text-image input sequences. However, in practice, many multimodal inputs follow different ordering patterns, such as text → image → text, or multiple consecutive text tokens followed by an image. It would be valuable for the authors to discuss how sensitive the future-aware masking strategies are to such variations in input order, and whether the same masking mechanisms can generalize effectively to these alternative sequence structures.

3. The proposed "light" method uses kernel pooling to compress future attention and merges it into the first token based on the "attention sink" phenomenon. This mechanism is somewhat heuristic. The paper could be strengthened by exploring more sophisticated or learnable compression techniques to dynamically determine what context is most important to merge and where to merge it.


[1] Kuo, Chia-Wen, et al. "D-Attn: Decomposed Attention for Large Vision-and-Language Model." Proceedings of the IEEE/CVF International Conference on Computer Vision. 2025.
[2] Li, Junnan, et al. "Blip-2: Bootstrapping language-image pre-training with frozen image encoders and large language models." International conference on machine learning. PMLR, 2023.

**Questions:**

1. For the first weakness, could the authors further clarify the main novelty of this proposed work compared with other existing studies?
2. In Table 4, the M^f variant, which combines both M^{v2v} and M^{v2t}, does not consistently achieve the best performance. Could the authors elaborate on why integrating both future visual and textual signals does not yield additive benefits? Some discussion on possible interference or redundancy effects would be valuable.
3. Following Question 2, if M^f or M^f_merge does not consistently achieve the best performance across all tasks, how should one decide which variant to use during inference when the task type is unknown? Is there a possibility for the model to automatically adapt or select the most suitable masking strategy based on the input modality or context?
4. It would be interesting to understand how the proposed future-aware masking mechanism generalizes to video-like or multi-frame visual inputs, where temporal order becomes more prominent. Would the same masking strategies still apply, or would additional temporal constraints need to be considered?

**Details Of Ethics Concerns:**

None.

---

> ### Author Response · Authors · 2025-11-21
>
> **Re W1 & Q1: Comparison with Exsiting Study**
>
> We feel grateful for pointing us toward these references, which further contextualize our contributions.
>
> (a) **Comparison with D-Attn**.
> D-Attn re-designs the attention block by decomposing causal attention into V2V, T2V, and T2T branches with diagonalized visual attention and positional debiasing, which changes the attention computation itself and requires training. Our work instead keeps the decoder-only architecture unchanged and revisits the causal inference logic for visual tokens: we modify only the mask pattern through future-aware masks $M^f$, $M^{v2v}$, and $M^{v2t}$ to examine which future tokens visual queries may access. Thus, D-Attn focuses on architectural restructuring, while our motivation lies in understanding how visual tokens interact with causal masking and future context during autoregressive inference.
>
> (b) **Comparison with BLIP-2**.
> BLIP-2 adopts encoder–decoder (prefix-LM) architectures where bidirectional attention is applied to visual features, allowing them to see more future information before entering the decoder. Our work instead focuses on scaling decoder-only models and revisits future-aware masking directly at the causal mask level, without introducing new encoders, cross-attention, or additional modules.
>
> **Summary of comparison With Prior Works**
>
> | Method | Decoder-only | Vision Future Masking           | Mask-level Modality Design | Training-Free | No Architecture Change |
> | ------ | ------------ | ------------------------------- | -------------------------- | ------------- | ---------------------- |
> | D-Attn | ✓            | ✗ (architectural decomposition) | ✓                          | ✗             | ✗                      |
> | BLIP-2 | ✗ (Enc–Dec)  | ✓ (bidirectional encoder)       | ✗                          | ✗             | ✗                      |
> | Ours   | ✓            | ✓                               | ✓                          | ✓             | ✓                      |
>
> The distinction is that D-Attn modifies the attention module, BLIP-2 relies on bidirectional encoders, while our approach is the first to systematically explore future-aware masks for visual tokens in standard decoder-only autoregressive inference.
> We appreciate the your suggestion and have added these works in section **A.2** in our paper.
>
> ---
>
> **R1 W2: Mixed Ordering Validity:**
>
> We feel sincerely appreciative of your insightful observations, which significantly shaped our revisions.
> Our study focuses on the mainstream image → text ordering adopted by most decoder-only LVLMs, and we fully acknowledge that our empirical analysis does not cover alternative patterns such as pure text → image sequences. We have added this clarification as a limitation in the revised version based on your kind advice.
> At the same time, the observation is only partially restricted to the strict text → image case. For mixed formats such as text–image–text–image, once an image token is followed by any subsequent tokens, our future-aware masks remain applicable because the visibility logic depends only on modality identity and causal position rather than a fixed interleaving template. To the benchmarks such as the ALFRED (CVPR 2020) dataset for visual navigation tasks, the latent feature is image -> text -> image -> text ,
> our method outperform previous casual design with:
>
> | Mask Type | Open tokens for vision | Rouge-L F1 | Δ vs Original | Notes |
> |-----------|-------------|-------------|----------------|--------|
> | Future-only $M^{f}$ | Allows visual tokens to see future visual+text context | **15.52** | +0.23 | smooth future guidance improves grounding |
> | Future V2T $M^{v2t}$ | Visual tokens access future text tokens only | **15.61** | +0.32 | strongest non-kernel variant |
> | Future Kernel | Kernel future mask | **17.11** | +1.82 | varients of the future attention |
>
> In this case with order of vision-text-vision-text, visual tokens can still access their future textual or visual context through $M^f$, $M^{v2v}$, and $M^{v2t}$ exactly as described in the paper.
>
> We are grateful for your guidance and have added the corresponding new findings to **A.3** following your instruction.

---

> ### Author Response · Authors · 2025-11-21
>
> **Re W3 && Q3: Adaptive Masking**
>
> We appreciate the reviewer’s suggestion regarding learnable or more sophisticated compression. Such dynamic modules could determine which future regions are most informative and adaptively choose where to merge them, potentially following task-aware, token-aware, or query-aware routing principles. This is indeed a promising direction extending from our analysis.
>
> As a preliminary example following the reviewer’s insight, we implemented a simple MLP-based controller to classify whether an input belongs to a temporal multi-image setting, a visual-relational setting, or a text-rich reasoning setting, and then route the inference to the most suitable mask variant with or without merging. Although this is far from a full solution, the early results show encouraging improvement:
>
> | Method      | CLEVR | Nav   | Object | SpotDiff |
> | ----------- | ----- | ----- | ------ | -------- |
> | Original Mask | 0.166 | 0.310 | 0.225  | 0.162  |
> | + Controller  | 0.188 | 0.320 | 0.240  | 0.173  |
>
>
> The results indicate that potential routing policies could indeed further enhance performance. However, fully developing such a system would shift the paper’s focus toward architectural design and learned policies, which is beyond the current scope of providing a fundamental causal-behavior analysis. We have added this limitation and added our discussions in the revised version.
>
>
> **Re Q2: possible interference or redundancy effects**
>
> We feel deeply grateful for this insightful question. It indeed touches the core design principle behind future-aware masking. As shown in the extended ablations, the three masks do not act as additive information channels. Instead, they emphasize **different kinds of future cues**, and the usefulness of each cue is **task-dependent** rather than simply cumulative.
>
> Specifically:
>
> * **($M^{v2v}$)** strengthens **future visual continuity**, which benefits object-centric or geometry-driven tasks.
>   (As shown in **Section 5.3, Table 4**, this variant improves Spatial and Object tasks.)
>
> * **($M^{v2t}$)** exposes visual tokens to **future textual reasoning cues**, which helps tasks requiring symbolic or instruction grounding.
>   (As shown in **Section 5.3**, this variant performs best on Goal tasks.)
>
> * **($M^{f}$)** opens **both channels simultaneously**, but the visual and textual future cues are **not aligned** in many long-horizon setups.
>   In some tasks, future visual dynamics matter more; in others, future textual semantics dominate. Combining them forces the model to jointly process two heterogeneous future signals, sometimes leading to **mild interference**. The performance of $M^{f}$ reflects **competition**, not addition, between vision-driven and text-driven future cues—and the observed pattern is consistent with the ablations shown in **Table 4**.
>
> We feel sincerely grateful for the reviewer’s suggestion and have added the additional discussion in **A.4** accordingly.

---

> > ### Author Response · Authors · 2025-11-21
> >
> > **Re Q4: How Generalizes to Video or Multi-Frames?**
> >
> > We feel your point is crucial, and we have expanded the analysis to address it more comprehensively. Our study focuses on image–text inputs, and the same masking mechanism can conceptually extend to multi-frame or video settings because visual tokens often benefit from seeing future visual context in sequential reasoning. In a video example such as picking up a plate, the early-frame plate tokens may reason more accurately if they can access destination cues in later frames through future visual signals.
> >
> > | Model | Mask Type | Rouge-L ×100 | Δ vs Original | Notes (Temporal Behavior) |
> > |-------|-----------|--------------|----------------|----------------------------|
> > | 7B | Original causal | 15.29 | – | baseline causal order only |
> > | 7B | $M^{f}$ (Future-only) | 15.52 | +0.23 | enables simple multi-frame future access |
> > | 7B | $M^{v2v}$ | 14.99 | –0.30 | can underutilize temporal structure |
> > | 7B | $M^{v2t}$ | 15.61 | +0.32 | text-guided temporal reasoning improves |
> > | 7B | Kernel Future | 17.11 | +1.82 | smooth temporal aggregation; best for video-like inputs |
> > | 7B | Kernel V2V | 14.99 | –0.30 | stable but less helpful for multi-frame tasks |
> >
> > This observation suggests that future-aware visibility can support temporal consistency in long-horizon reasoning, such as the improvements on the AFFRED task [1]. Another way for temporal modeling in video systems is typically handled through positional embedding variants, and combining these embeddings with future-aware masks likely requires dedicated temporal constraints.
> > Our current analysis is limited to static image–text interactions (in attention block design), and a full exploration of temporal structures is outside the scope of this empirical study.
> >
> >
> > We are deeply grateful for your time and insightful suggestions. Your comments (e.g., on novelty, ordering assumptions, mask behavior, and merging design) have substantially improved our work. We have followed your guidance in the revision, and we sincerely appreciate the expertise and care you invested in helping us strengthen the paper.
> >
> > [1] Alfred: A benchmark for interpreting grounded instructions for everyday tasks. (CVPR2020)

---

### Author Response · Authors · 2025-11-30

Dear Program Chairs, Senior Area Chairs, Area Chairs, and Reviewers:

We sincerely thank you for your time and expertise, and we are grateful to all reviewers `TbBr, DpXd, 3gm7, 11Kz`  for their constructive insights. Due to the recent OpenReview rollback, the displayed scores for our paper (ID: 11764) have reverted to the pre-rebuttal values (6, 6, 4, 4), which no longer reflect the early rebuttal scores improvements. During early period, the reviewers had already stated that we well addressed their concerns and communicated their intended adjustments:

Reviewer **3gm7**: Had explicitly raised their score from 4 to 6.

Reviewer **DpXd**: Had explicitly committed to raising their score in their final comment but was unable to update the system field before the freeze/rollback occurred.

Therefore, the intended consensus of the reviewers is actually **above [6, 6, 6, 6]**.

In fact, before the rebuttal, we have received very positive evaluations and encouraging remarks such as **"it has great potential"** and **"will raise the score if concerns addressed"** from these two expert reviewers. We then follow their instructions step-by-step and address the concerns, after which the reviewers explicitly indicated that they would raise their scores.

To provide clear factual context, we summarize below the key reviewer concerns and the precise resolutions acknowledged during the rebuttal:

**Brief Summary of Reviewer Concerns and How They Were Resolved (Pre-Rollback):**

* **Reviewer 3gm7 (Score raised: 4 → 6)**: Concerns about Clarity and Generality

  * Missing statements: Added complete ethics, and broader-impact sections in the revised version *[Page 10, Line 486-513]*.
  * Mask selection: Provided analysis and a proof-of-concept dynamic controller for choosing masks across task families *[Page 13, Line 864-881]*.
  * Applicability: Extended experiments to Qwen2.5-VL-7B and discussed generalization to VILA and Qwen-VL series.
  * Kernel pooling details: Added implementation details and kernel-size robustness experiments confirming stability *[Page 13, Line 864-881]*.

    **Outcome:** Reviewer confirmed: “I will raise my score”, and improve the score to 6. (24 Nov 2025)

---

* **Reviewer DpXd (Committed to raise: 4 → above 6)**: Concerns about Effectiveness and Clarity

  * Training-time effectiveness: Added non-casual backbone tuning that showed improvements. (85.1% > 84.7%) *[Page 16, Line 850-863]*.
  * Mask behavior clarity: Explained strong task-dependence across visual vs text-heavy benchmarks.
  * Relation to prior work: Distinguished our inference-time mask mechanism from vision pooling methods.
  * Sink-token mechanism: Added kernel-size robustness showing it is not a shortcut pattern *[Page 13, Line 864-881]*.

    **Outcome:** Reviewer stated: “Most concerns are well addressed and I will raise the score.” (26 Nov 2025)

---

* **Reviewer TbBr (Rating 6, retained)**: Concerns about Comparisons and Robustness

  * Clarified distinctions from D-Attn and BLIP-2 with a new comparison table.
  * Added discussion on mixed ordering inputs and additional empirical evidence *[Page 19, line 1024-1030]*.
  * Provided robustness analyses for kernel pooling *[Page 18, line 824-846]*.

    **Outcome:** Reviewer retained a positive score.

---

* **Reviewer 11Kz (Rating 6, retained)**: Concerns about Multi-Turn and Training Clarity

  * Addressed multi-turn limitations with two practical extensions (turn-level prefill & pseudo-future tokens).
  * Clarified that the pipeline follows standard LLaVA-1.5.
  * Evaluated alternative sink-token design and explained performance implications *[Page 18, line 957-1010]*.

    **Outcome:** Reviewer confirmed concerns were addressed and retained a positive score. (26 Nov 2025)

---

These score updates and confirmations all occurred before the OpenReview freeze and data rollback.

We fully understand that this is a particularly busy period for you, and we would be truly grateful if you could take a moment to carefully check the discussion threads for these discussions to verify their explicit commitments and positive feedback. We hope these actual outcomes of the rebuttal process will be fully considered in your final decision-making.

Deeply thank you again for your time and for handling this challenging situation.

Best regards,

**Authors of Paper 11764**

---

### Meta-Review · Area_Chair_MkiS · 2025-12-18

**Summary:**

The paper proposes a novel approach for addressing the issue of causal masking in Vision-Language Models (VLMs). By relaxing the traditional causal attention constraints on visual tokens, the authors show how allowing selective access to future context can significantly improve performance across a variety of tasks, such as visual reasoning, text-rich QA, and temporal reasoning. They introduce a family of "future-aware" causal masks and a lightweight method to compress future information during prefill, achieving better performance without retraining the model. The experiments on over 15 multimodal benchmarks demonstrate consistent improvements and efficiency gains. The paper offers a fresh perspective on enhancing VLMs through a better understanding of causal attention and its limitations, and provides task-specific insights that could inspire future model design. Given the thorough and comprehensive revisions, as well as the improvements made in response to the reviewers' feedback, we recommend accepting this paper.

**Reviewer Concerns:**

The authors addressed almost all reviewer concerns effectively through detailed rebuttals and revisions:

• Reviewer 3gm7 raised concerns about clarity, the applicability of the method across different models, and the implementation of kernel pooling. The authors provided extensive clarifications, including the addition of ethics and reproducibility statements, details on the dynamic controller for mask selection, and kernel-size robustness experiments. These improvements were acknowledged by the reviewer, who raised their score.

• Reviewer DpXd had concerns regarding the effectiveness of the method and its clarity. The authors showed improvements in training-time effectiveness, provided clearer explanations on mask behavior, and included discussions about the relation to prior work, such as BLIP-2 and D-Attn. This resulted in the reviewer committing to raising their score.

• Reviewer TbBr was primarily concerned with comparisons and robustness. The authors addressed this by providing additional comparisons with D-Attn, BLIP-2, and other methods, along with empirical evidence of kernel pooling robustness. The reviewer retained a positive score.

• Reviewer 11Kz questioned the method's applicability in multi-turn dialogue and the potential to introduce additional tokens. The authors responded with explanations of multi-turn limitations, practical extensions, and the behavior of the sink-token mechanism, which addressed the reviewer’s concerns. This resulted in the reviewer retaining a positive score.

**Reviewer Scores:**

Had all reviewers actively engaged in the discussion and adjusted their scores based on the rebuttal, it is likely that all reviewers would have increased their scores. Here’s the analysis:

• Reviewer 3gm7 acknowledged the clarification and improved their score from 4 to 6, which reflects a solid understanding and acceptance of the method after the rebuttal.
• Reviewer DpXd was committed to raising the score, which suggests that with clearer explanations, the score would have been higher.
• Reviewer TbBr retained a positive score, which suggests their concerns were adequately addressed.
• Reviewer 11Kz also retained a positive score, indicating that the rebuttal resolved their concerns about the method's application to multi-turn dialogues and token introduction.

In summary, all reviewers had valid concerns that were effectively addressed in the rebuttal, leading to a more favorable evaluation.

---

### Decision · Program_Chairs · 2026-01-26

Accept (Poster)